# Planar cell polarity-mediated induction of neural stem cell expansion during axolotl spinal cord regeneration

Aida Rodrigo Albors[1,2,3†], Akira Tazaki[1,2,3†], Fabian Rost[4], Sergej Nowoshilow[1,2,3], Osvaldo Chara[4,5], Elly M Tanaka[1,2,3*]

[1]Deutsche Forschungsgemeinschaft – Center for Regenerative Therapies Dresden, Dresden, Germany; [2]Max Planck Institute of Molecular Cell Biology and Genetics, Dresden, Germany; [3]Technische Universität Dresden, Dresden, Germany; [4]Center for Information Services and High Performance Computing, Technische Universität Dresden, Dresden, Germany; [5]Institute of Physics of Liquids and Biological Systems, National Scientific and Technical Research Council, University of La Plata, La Plata, Argentina

**Abstract** Axolotls are uniquely able to mobilize neural stem cells to regenerate all missing regions of the spinal cord. How a neural stem cell under homeostasis converts after injury to a highly regenerative cell remains unknown. Here, we show that during regeneration, axolotl neural stem cells repress neurogenic genes and reactivate a transcriptional program similar to embryonic neuroepithelial cells. This dedifferentiation includes the acquisition of rapid cell cycles, the switch from neurogenic to proliferative divisions, and the re-expression of planar cell polarity (PCP) pathway components. We show that PCP induction is essential to reorient mitotic spindles along the anterior-posterior axis of elongation, and orthogonal to the cell apical-basal axis. Disruption of this property results in premature neurogenesis and halts regeneration. Our findings reveal a key role for PCP in coordinating the morphogenesis of spinal cord outgrowth with the switch from a homeostatic to a regenerative stem cell that restores missing tissue.

*For correspondence: elly.tanaka@crt-dresden.de

†These authors contributed equally to this work

**Competing interests:** The authors declare that no competing interests exist.

## Introduction

An outstanding challenge in biology is to understand how resident stem cells are mobilized to regenerate injured or missing tissues. During regeneration the removal of tissue activates a program that must coordinately control (i) the rate of cell divisions, (ii) the type of daughter cells generated, and (iii) the spatial rearrangements of these cells within the tissue.

Axolotl spinal cord regeneration is an ideal context in which to link molecular signalling with the control of stem cell properties since the stem cell population is clearly identifiable and manipulable. Cells with stem cell features persist throughout life lining the spinal cord central canal of the axolotl. These cells divide occasionally to add neurons to the intact central nervous system of the axolotl, presumably by asymmetric divisions (*Holder et al., 1991*). Following tail amputation, however, the pool of SOX2[+] neural stem cells located within 500 μm of the injury is mobilized to regenerate the missing portion of the spinal cord (*Fei et al., 2014*; *McHedlishvili et al., 2007*). First, these SOX2[+] cells proliferate extensively, expanding as a tube along the anterior-posterior (AP) axis in concert with overall tail regeneration (*Egar and Singer, 1972*; *Holtzer, 1956*; *McHedlishvili et al., 2007*; *Nordlander and Singer, 1978*). Although a few neurons from the source zone move and integrate into the regenerating cord, their contribution is minimal as they do not seem to dedifferentiate or divide during the process (*Zhang et al., 2003*). It is later on, when the SOX2[+] cells closest to the

**eLife digest** Stem cells found in adult tissues are vitally important for tissue repair and maintenance. These cells divide in two main ways: equally to create two new stem cells, or unequally to create a stem cell and a cell that can develop into one of the cell types in the tissue. A key challenge for biologists is to understand how these tissue-resident stem cells are activated and organized to regenerate injured or missing tissue.

Throughout the life of the axolotl salamander, neural stem cells in the spinal cord occasionally divide to add new nerve cells to the healthy spinal cord. However, the axolotl can also regenerate part of its spinal cord, for example if its tail is lost. Under these conditions, the neural stem cells can convert into a highly regenerative stem cell that can produce all the different cell types needed to regrow the spinal cord.

As a stem cell becomes a new cell type, it activates different sets of genes. Therefore, Rodrigo Albors, Tazaki et al. measured gene activity in the neural stem cells involved in axolotl spinal cord regeneration to uncover how these cells develop into a more regenerative form. This revealed that when an axolotl tail is amputated, resident stem cells turn off the genes that are specifically active in neuron-generating cells. In addition, they activate a similar set of genes to that seen in the embryonic cells that form the developing nervous system. These genes speed up cell division and activate an important signaling pathway.

This pathway – the Wnt/PCP pathway – fulfils various developmental roles, one being to orient cell divisions, particularly in elongating tissues. In axolotls, this pathway causes the stem cells to divide equally to increase the number of available stem cells, and orients the direction of these divisions to ensure that the regenerating spinal cord elongates correctly. If this pathway is disrupted, the cells return to dividing unequally, generating nerve cells prematurely and halting the growth of the spinal cord. Such insights could help develop methods of repairing damaged nervous tissue in other animals that cannot regenerate to the extent that axolotls can.

amputation plane resume neurogenesis that the differentiated cell types of the new spinal cord are gradually replenished in a head-to-tail sequence while cells at the posterior end continue rapid expansion (*Holtzer, 1956*; *McHedlishvili et al., 2007*). Importantly, lineage tracing of cells lining the central canal showed that at least some cells in the source zone become highly proliferative and multipotent, populating different dorsal-ventral regions of the spinal cord (*McHedlishvili et al., 2007*; *McHedlishvili et al., 2012*). Furthermore, clonal isolates, when transplanted into the regenerating spinal cord, contributed to neurons, astrocytes, as well as oligodendrocytes in all regions of the spinal cord, indicating that a single cell can give rise to all cells of the spinal cord (*McHedlishvili et al., 2012*). Thus, cells in the axolotl spinal cord display features of true stem cells, since they can self renew and give rise to all the differentiated cell types of the new spinal cord (*Echeverri and Tanaka, 2002*; *McHedlishvili et al., 2007*; *McHedlishvili et al., 2012*). How a stem cell that is neurogenic under homeostasis converts upon tail amputation to a highly regenerative stem cell remains unknown.

Since tail regeneration represents restoration of the primary body axis, an interesting further question is whether resident spinal cord cells reacquire embryonic properties to achieve this switch to a regenerative state. During spinal cord development, SOX2[+] cells residing in the so-called stem zone (SZ) progressively coalesce to form the neural plate (pre-neural tube, PNT) that then closes into the neural tube (*Wilson et al., 2009*). Although PNT cells express *Sox2* they do not yet express neuronal transcription factors and thus, remain multipotent and proliferating (*del Corral et al., 2003*; *del Corral and Storey, 2004*). Cells in the neural tube acquire neural progenitor identity as they start expressing neuronal transcription factors and commit to produce the cell types of the adult spinal cord (*del Corral et al., 2003*; *Jessell, 2000*). Whether the neural stem cells in the adult axolotl spinal cord revert to a state resembling one of these developmental stages to rebuild the spinal cord is not known.

Here, we show that tail amputation in the axolotl causes resident spinal cord stem cells to reactivate an embryonic-like gene expression program associated with proliferative, multipotent

neuroepithelial cells that undergo axis elongation. A critical part of this program is the reactivation of Wnt/planar cell polarity (PCP) signalling precisely within the cells that will regenerate the new spinal cord. Investigation of this pathway during regeneration revealed that PCP simultaneously controlled posteriorward orientation of cell divisions and the switch from neurogenic divisions to those divisions that expanded the stem cell pool. Together, these findings provide new insights into how molecular cues initiated by injury control the cell biology of neural stem cells to yield complete spinal cord regeneration in the axolotl.

## Results

### Neural stem cells in the injured axolotl spinal cord reactivate molecular programs associated with embryonic neuroepithelial cells

Although the regenerating tail shows morphological differences to the developing embryonic axis, the requirement to produce new regions of the spinal cord raised the possibility that developmental factors controlling spinal cord development are reactivated during regeneration. To establish whether regenerating axolotl neural stem cells dedifferentiate to an embryonic-like state we referred to expression profiling data of chick neural development that exploited the developmental gradient along the neuraxis to profile samples corresponding to the stem zone (SZ), pre-neural tube (PNT), caudal (CNT) and rostral neural tube (RNT) (*Olivera-Martinez et al., 2014*). To investigate the transcriptional profile of regenerating versus homeostatic axolotl neural stem cells we focused on axolotl orthologs to the 100 chicken genes that changed most significantly at the onset of neurogenesis, as captured in the pooled SZ+PNT and CNT+RNT comparison (50 upregulated and 50 downregulated genes) (*Olivera-Martinez et al., 2014*). Specifically, we isolated RNA from the uninjured spinal cord (day 0), the 500 µm source zone 1 day after amputation (day 1), and the regenerating spinal cord 6 days after amputation (day 6), and used NanoString technology (*Geiss et al., 2008*) to measure transcript levels of the 100-gene set (*Figure 1A*). Differential expression analysis between regenerating and uninjured samples showed that most of the transcripts that are differentially regulated during development undergo significant regulation during regeneration (*Figure 1B* and *Figure 1—source data 1*). Direct comparison of changes in gene expression between datasets showed that 37 out of 50 chick genes low in the SZ+PNT versus CNT+RNT are downregulated in day 1 or day 6 axolotl samples compared to day 0, and 18 out of 50 chick genes high in the SZ+PNT versus CNT+RNT are upregulated in day 1 or day 6 axolotl samples (significant association, $p$=0.001294 as result of Fisher's exact test on count data) (*Figure 1C*).

Highly upregulated genes common to the SZ+PNT and the regenerating samples included the early growth response gene *Egr1* (*Milbrandt, 1987*) and the estrogen-induced *Greb1* (*Ghosh et al., 2000*), both associated with growth regulation (*Liao et al., 2004*; *Rae et al., 2005*); and the caudal gene *Cdx4*, a key player in embryonic body axis elongation (*Faas and Isaacs, 2009*; *Shimizu et al., 2005*; *van Nes et al., 2006*). Conversely, a number of neuronal transcription factors that were higher in the neurogenic CNT+RNT than the SZ+PNT zone were downregulated in the regenerating spinal cord (*Pax6, Pax7, Tfap2b, Prdm12 Irx1, Nr2f2, Dbx2, Zic1* and *Hes6*), along with genes related to signaling pathways that during embryonic development promote neurogenesis such as the retinoic acid signaling-readout genes *Crabp1* and *Rarb* (*del Corral et al., 2003*).

The biological outcome of many signaling pathways varies depending on the cellular context, and the type of receptor expressed. Intriguingly, the expression of the bone morphogenetic protein (BMP) receptor *Bmpr1b* and the fibroblast growth factor (FGF) receptor *Fgfr2*, both linked to neurogenic differentiation during development and as such more highly expressed in the CNT+RNT than in the SZ+PNT (*Panchision et al., 2001*, *Zhang et al., 2002*), were downregulated in the regenerating spinal cord compared to the uninjured spinal cord. Likewise, the expression patterns of Eph receptors in the proliferative SZ+PNT seem to be recapitulated in the regenerating spinal cord with the upregulation of EphA1 receptor and continued, although lower, expression of EphA5 in comparison with the uninjured spinal cord. The Eph family of receptor tyrosine kinases have been implicated among others in regulating cell proliferation and differentiation of neural progenitors, neuron survival, axon regrowth and axon sprouting, in the developing and adult nervous system (*Conover et al., 2000*; *Khodosevich et al., 2011*; *North et al., 2009*). Their activities depend on the combination of Eph receptor and ephrins (Eph receptor interacting proteins) (*Dodelet and*

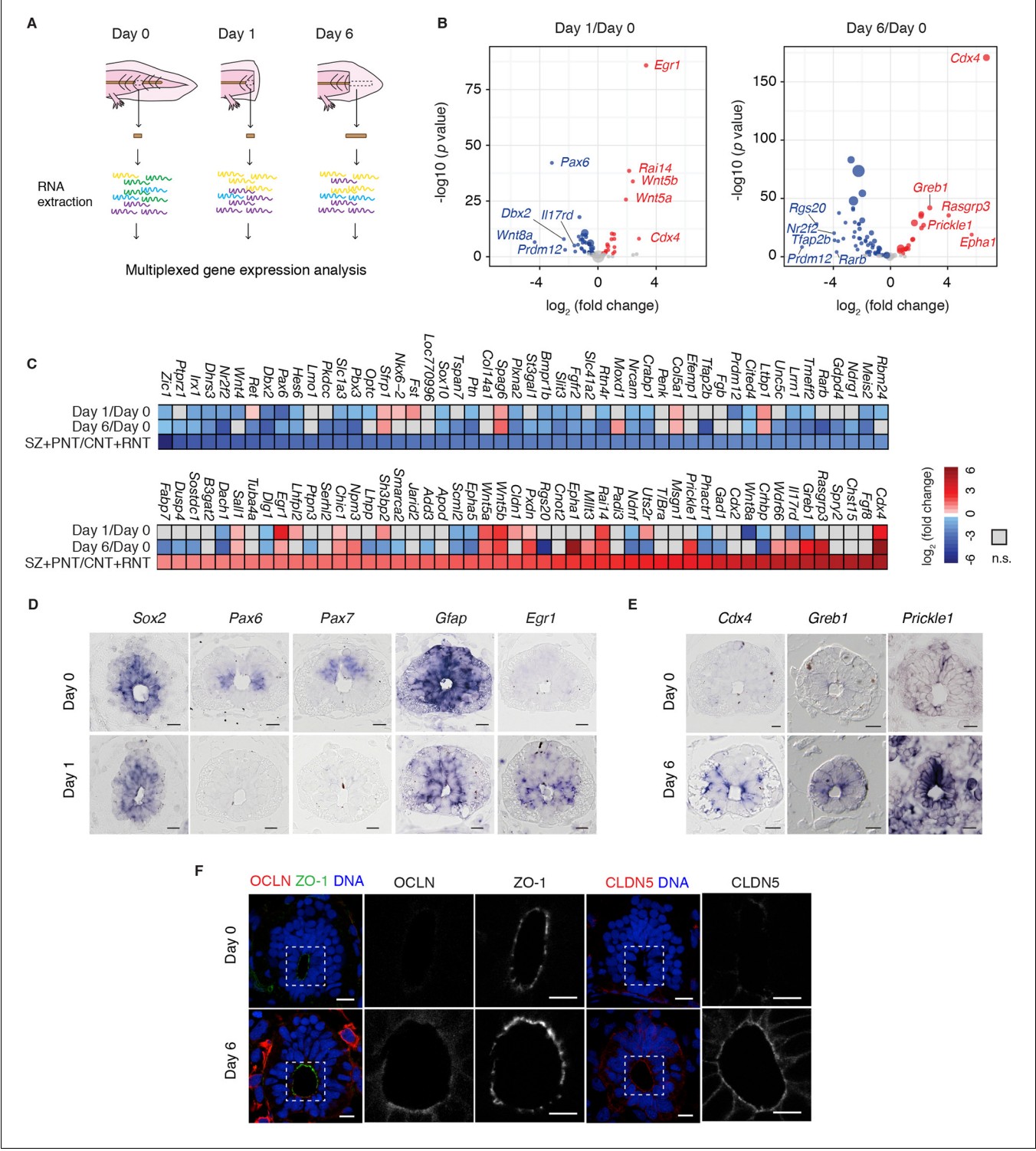

**Figure 1.** Neural stem cells in the axolotl spinal cord reactivate molecular features associated with embryonic neuroepithelial cells upon injury. (**A**) Schematic of the distinct regions microdissected for NanoString expression profiling. 48–50 pieces of spinal cord were pooled together for each of the 3 biological replicates per group. (**B**) Changes in gene expression ($\log_2$ fold change) in cells of the source zone at day 1 (left) and regenerating spinal cord at day 6 (right) versus cells in the uninjured spinal cord (day 0). Y-axis shows the statistical significance of these changes ($-\log_{10} p$ value). Red dots, upregulated genes; blue dots, downregulated genes; grey dots, genes that are not significantly regulated in regenerating versus uninjured samples. N=3 samples per group. *P* values indicate the results of negative binomial tests. $p < 0.05$ (FDR < 0.07) was considered significant. Dot area is proportional to the number of transcripts counted for that gene. Names are the top five upregulated (red) and top five downregulated (blue) genes in

*Figure 1 continued on next page*

*Figure 1 continued*

the dataset. (C) Heat map of fold change in gene expression values in regenerating spinal cord versus uninjured spinal cord, side by side with the reported fold change values in the stem zone and pre-neural tube (SZ+PNT) compared to more advanced rostral and caudal neural tube (CNT+RNT) of the embryonic chick spinal cord, as described in (*Olivera-Martinez et al., 2014*). Dataset used to generate the volcano plots and heat maps can be found in *Figure 1—source data 1*. (D) In situ hybridization on spinal cord cross-sections at day 0 (top) and of the regeneration source zone at day 1 (bottom). (E) In situ hybridization on spinal cord cross-sections at day 0 (top) and regenerating spinal cord at day 6 (bottom). (F) Immunofluorescence on cross-sections of uninjured (top) and regenerating spinal cord at day 6 (bottom) for the apical junctional proteins ZO-1, OCLN, and CLDN5. DNA is labeled with Hoechst. Insets show the apical surface of the spinal cord at higher magnification. Images are single optical sections using the same acquisition parameters on sections from the same slide. Observations were consistent in more than 5 axolotls. Scale bars, 20 μm.

The following source data is available for figure 1:

**Source data 1** Gene expression changes in the regenerating axolotl spinal cord compared to the uninjured axolotl spinal cord.

*Pasquale, 2000*; *Lisabeth et al., 2013*). These changes are consistent with a switch from pro-neurogenic to pro-proliferative signaling modalities in the regenerating spinal cord.

In that line, mitogen-activated protein kinase (MAPK) signaling is presumably more active in the regenerating spinal cord compared to the uninjured spinal cord, based on the strong upregulation of its transcriptional target gene *Rasgrp3* and downregulation of the negative regulator *Dusp4*. It is perhaps surprising that we could not detect transcripts for FGF8, which activates this pathway in the developing spinal cord. However, although other FGFs were not present in the 100-gene list, FGF2 has been shown to be upregulated in the regenerating spinal cord of in another salamander species and thus could be activating MAPK signaling in the context of spinal cord regeneration (*Zhang et al., 2000*).

Although a more extensive analysis of signalling activities during axolotl spinal cord regeneration will require a global genome-wide approach, the activation of non-canonical Wnt/PCP signaling during regeneration, reflected in the strong upregulation of *Wnt5a, Wnt5b, Prickle1* genes, was notable in our dataset. These genes are also highly upregulated in the developmental SZ+PNT sample, consistent with the notion that regenerating neural stem cells in the axolotl dedifferentiate to a SZ+PNT-like state.

To confirm that the gene expression changes occurred within the neural stem cells we performed in situ hybridization of selected factors. While transcript levels of the neural stem cell marker *Sox2* did not visibly change during regeneration, the neuronal transcription factors *Pax6* and *Pax7* and the glial marker *Gfap* showed indeed lower expression levels in *Sox2*[+] cells at day 1 (*Figure 1D*). Conversely, examination of upregulated genes such as *Egr1* at day 1 and *Cdx4, Greb1*, and PCP gene *Prickle1* at day 6 showed an induction of signal in *Sox2*[+] cells of regenerating samples (*Figure 1E*). These expression results were consistent with the mature SOX2[+] stem cells dedifferentiating during regeneration to a developmental state similar to the SZ+PNT.

A distinctive characteristic of embryonic PNT neuroepithelium is its marked apical-basal polarity including the expression of tight junction proteins (*Aaku-Saraste et al., 1996*). To extend the findings above, we immunostained axolotl spinal cord cross-sections for the tight junction proteins occludin (OCLN) and claudin-5 (CLDN5), previously shown to be expressed in PNT cells but downregulated at the onset of neurogenesis (*Aaku-Saraste et al., 1996*). We observed strong OCLN and CLDN5 signal in the apical side of regenerating neural stem cells compared to cells in the uninjured spinal cord, while the junctional protein ZO-1 did not change (*Figure 1F*). Taken together, these findings suggest that the regenerative axolotl neural stem cells acquire canonical features of embryonic neuroepithelial cells of the PNT including the onset of factors associated with growth regulation such as *Egr1*, axis elongation such as *Cdx4, Wnt5a* and *Wnt5b*, as well as epithelial organization such as OCLN and CLDN5.

## Neural stem cells transit to a rapid, proliferative cell division mode during regeneration

A functional characteristic of embryonic neural progenitors is the lengthening of their cell cycle as they switch from symmetric, proliferative divisions to asymmetric, neurogenic divisions (*Calegari et al., 2005*; *Olivera-Martinez et al., 2014*; *Takahashi et al., 1995*; *Wilcock et al., 2007*).

We therefore asked whether the dedifferentiation of axolotl neural stem cells extended to the acquisition of a rapid, proliferative mode of division. The exponential-like growth of the spinal cord during the first 8 days after amputation indeed seemed to support this idea (*Figure 2A*).

First, to determine the fraction of proliferative neural stem cells in uninjured and regenerating settings we immunostained tissue sections for SOX2 and the proliferation marker proliferating cell nuclear antigen (PCNA) (*Figure 2B*). The difference in the fraction of PCNA$^+$ cells was surprisingly small between the non-regenerating and regenerating spinal cord (*Figure 2C*). Mitotic index quantifications (see Materials and methods), however, revealed that a low percentage of these SOX2$^+$-PCNA$^+$ cells in the uninjured spinal cord are mitotic (*Figure 2D*, day 0). In contrast, we found a stable, threefold increase in mitotic figures in the source zone and regenerating spinal cord by day 4 (*Figure 2D*). These results suggest that neural stem cells proliferate slowly in homeostasis and accelerate their cell cycle during regeneration.

To investigate this possibility, we determined the cell cycle parameters of neural stem cells in the uninjured and regenerating spinal cord, in regenerating and non-regenerating populations, at the time of maximum expansion (starting at day 6) (*Figure 2E*). For that, we combined the above mitotic index measurements with a cumulative DNA labeling approach using the thymidine analog bromodeoxyuridine (BrdU) (*Nowakowski et al., 1989*). Briefly, we collected tail tissue at different times of continuous BrdU labeling and calculated the fraction of BrdU-labeled cells lining the central canal of the spinal cord. The fraction of BrdU-labeled cells increased over time as cells duplicated their DNA, until all proliferating cells were labeled (*Figure 2F*). In line with PCNA results, most cells in non-regenerative populations (0.80 ± 0.01 and 0.82 ± 0.01 in uninjured and non-regenerating region of the spinal cord, respectively) and almost all cells in the regenerating spinal cord took up BrdU (0.96 ± 0.01). However, the time needed to completely label the proliferating population of the regenerating spinal cord was much shorter (~31 h) than the time required for complete labeling of the uninjured (~192 h) and non-regenerating (~185 h) populations, suggesting that regenerating neural stem cells progress much faster through the cell cycle. By using a modeling approach (see Materials and methods), we calculated the average cell cycle length and the length of the different phases of the cell cycle for the different populations (*Figure 2G* and *Supplementary file 1*). These calculations indicated that uninjured and non-regenerating populations have an average cell cycle of 324 h ± 21 h and 339 h ± 32 h, respectively. In contrast, regenerating cells undergo significantly shorter cell cycles (119 h ± 10 h). These differences arise largely from a shorter G1 phase in regenerative cells compared to non-regenerative cells but also a shorter S phase.

To determine whether cells underwent symmetric, proliferative divisions or neurogenic divisions, we followed EdU$^+$ cells over time. To label cells undergoing DNA synthesis and their progeny we injected the thymidine analog ethynildeoxyuridine (EdU) daily from day 3 to day 8 after amputation. We then quantified the fraction of EdU$^+$ cells that were positive or not for the neural stem cell marker SOX2 in regenerating and non-regenerating regions of the spinal cord (*Figure 2H*). While in the non-regenerating region 84% ± 2% of the EdU$^+$ cells were SOX2$^+$, in the regenerating spinal cord virtually all EdU$^+$ cells were SOX2$^+$ (99% ± 1%) (*Figure 2H*). These observations indicated that whereas non-regenerating cells presumably exit the cell cycle to produce SOX2$^-$ neurons, regenerating cells undergo proliferative divisions at least during the first 8 days of regeneration. It should be noted that the difference in neurogenesis between the non-regenerating and the regenerating spinal cord is underestimated in this measurement paradigm due to the large difference in cell cycle lengths. Within the six day labeling period, all of the regenerating SOX2$^+$ cells were expected to complete at least one cell cycle, whereas, only 50% of the SOX2$^+$ cells would have completed a full cell cycle. Normalizing for this difference, we would expect that after a full cell cycle, only 70% of the EdU$^+$ cells in the non-regenerating portion would express SOX2. Together, these results indicate that neural stem cells speed up their cell cycle as they switch from a neurogenic to a proliferative mode of division that leads to the exponential-like growth of the regenerating spinal cord.

## *Wnt5a* and *Wnt5b* and the PCP member *Prickle1* are upregulated in the regeneration source zone and regenerating spinal cord

*Wnt5a*, *Wnt5b*, and the core PCP gene *Prickle1* (*Gray et al., 2011*) constituted three of the most upregulated genes in our regenerating spinal cord dataset (*Figure 1B* and *Figure 3A*). Because Wnts, like Wnt5a, can signal through the PCP pathway to drive tissue growth and morphogenesis during embryonic development (*Gao et al., 2011*; *Wallingford, 2012*), we set out to investigate

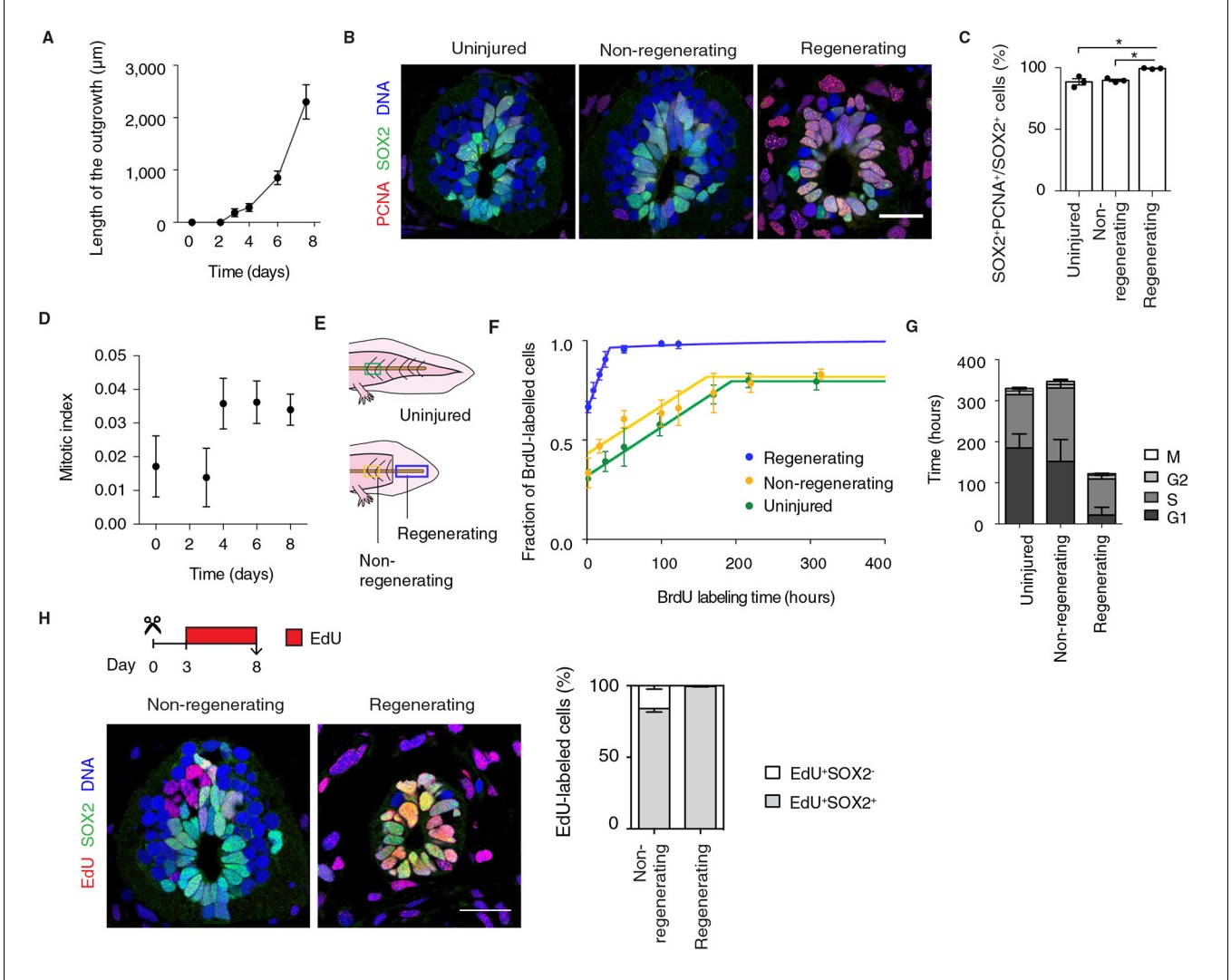

**Figure 2.** Fast, proliferative divisions underlie the expansion of neural stem cells in the regenerating spinal cord. (**A**) Length of spinal cord outgrowth during the first 8 days of regeneration. Data, mean ± s.d. of n=3 tails/time-point. (**B**) Immunofluorescence for the neural stem cell marker SOX2 and the proliferation marker PCNA on cross-sections of uninjured spinal cord and regenerating and non-regenerating regions of the spinal cord at day 6 after amputation. DNA is labeled with Hoechst. Scale bar, 50 µm. (**C**) Percentage of SOX2$^+$PCNA$^+$/SOX2$^+$ cells per cross-section in regenerative and non-regenerative settings, from images as in **B**. Error bars, mean ± s.e.m of n=3 tails/region. $P$=0.691, uninjured versus non-regenerating; $P$=0.014, uninjured versus regenerating; $p$=0.001, non-regenerating versus regenerating (two-tailed unpaired Student's $t$-test). Asterisks indicate statistical significance at $p$<0.05. (**D**) Mitotic index during the first 8 days of regeneration. Error bars, mean ± propagated errors of n=3 tails/time-point. (**E**) Schematic of the distinct cell populations considered in the cell cycle analysis. The uninjured neural stem cell population (green) was analyzed from uninjured tails; the non-regenerating cell population, from 1,500 µm anterior to the amputation plane (yellow); and the regenerating cell population from the regenerating spinal cord, posterior to the amputation plane (blue). (**F**) Fraction of BrdU-labeled cells at different times of continuous BrdU labeling. Curves are the least-squares fit of a mathematical model of BrdU incorporation to the experimental data (see Materials and methods). Colors indicate the neural stem cell populations as in *Figure 2E*. Error bars, mean ± s.d. of n≥5 tails/time-point. (**G**) Cell cycle parameters of uninjured, non-regenerating, and regenerating neural stem cell populations, calculated from data in D and F. Error bars, mean ± 1σ confidence intervals. Data can be found in *Supplementary file 1* and http://nbviewer.jupyter.org/gist/fabianrost84/3cc58a27b5688f4e2eba. (**H**) EdU was injected daily from day 3 to day 8 of regeneration, when tails were collected for analysis. Representative images of non-regenerating and regenerating regions of the spinal cord. SOX2 labels neural stem cells, EdU labels cells that underwent DNA synthesis and their progeny, and Hoechst labels DNA. Quantification of the percentage of EdU$^+$ cells that remain as neural stem cells (SOX2$^+$) or differentiatiate (SOX2$^-$) over total EdU$^+$ cells after the six-day chase. Error bars, mean ± s.e.m. of n=4 tails/region. Scale bar, 50 µm. A supplementary IPython notebook (*Pérez and Granger, 2007*) containing all the raw data and the code used for the estimations of mitotic index, cell cycle length and cell cycle parameters is available at http://nbviewer.jupyter.org/gist/fabianrost84/3cc58a27b5688f4e2eba.

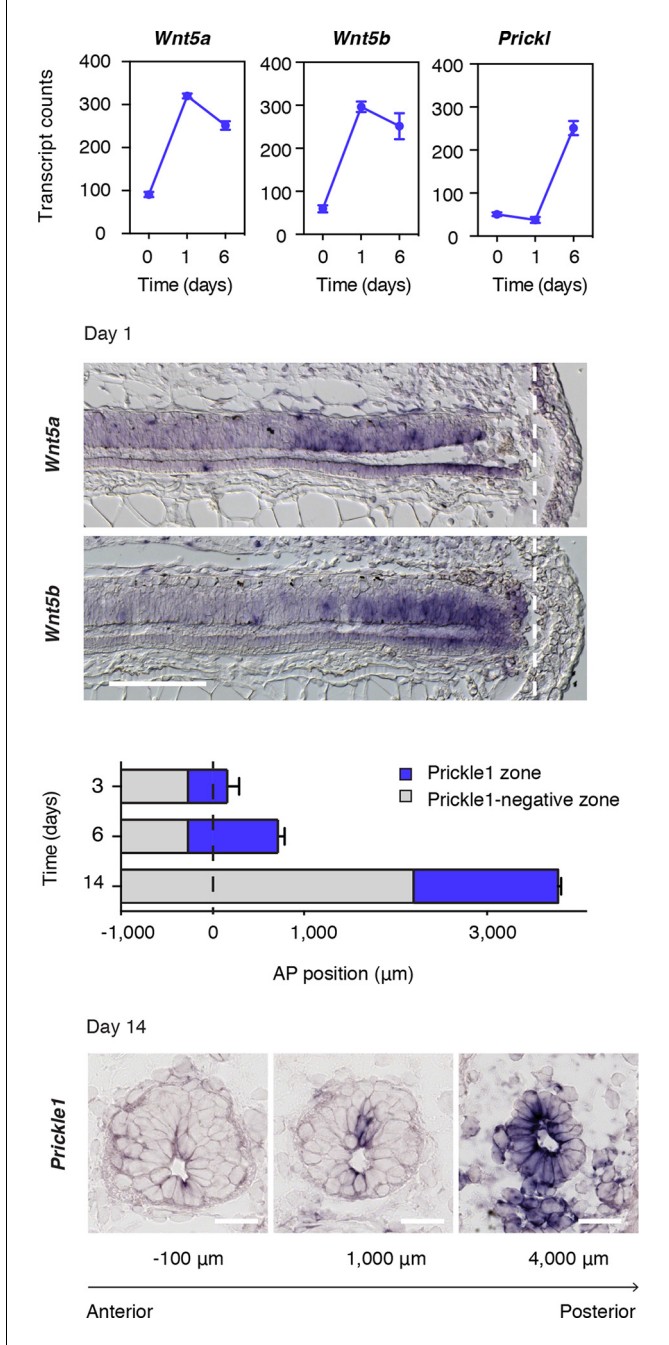

**Figure 3.** *Wnt5* genes and the PCP gene *Prickle1* are upregulated during spinal cord regeneration. (A) Mean ± s.d. of transcript counts of *Wnt5a, Wnt5b,* and *Prickle1* in uninjured and regenerating spinal cord as measured by NanoString (see also *Figure 1*). N=3 samples/time-point. (B) *Wnt5a* and *Wnt5b* in situ on longitudinal sections of axolotl tails 1 day after amputation. Note that *Wnt5a* and *Wnt5b* transcripts are precisely upregulated in the source zone. Dashed lines, amputation plane. Scale bar, 400 μm. (C) Quantification of the Prickle1 zone (blue boxes) on cross-sections along the AP axis of regenerating spinal cords at day 3 (429 μm ± 32 μm), day 6 (981 μm ± 74 μm), and day 14 (1579 μm ± 129 μm). Data, mean ± s.d. of n=3 tails/time-point. (D) Representative images of *Prickle1* in situ at different AP levels of a 14 days regenerating spinal cord. The numbers below each image indicate its AP position from the amputation plane. Negative numbers are anterior and positive numbers posterior to the amputation plane. Note that *Prickle1* is highly expressed in the caudal-most portion of the regenerating spinal cord that remains as a simple neuroepithelium but undetectable close to the amputation plane, in the more advanced spinal cord tissue. Scale bars, 50 μm.

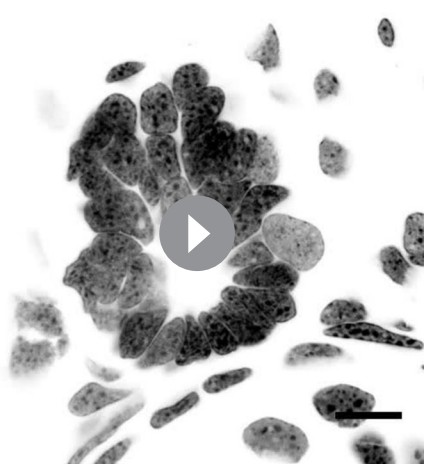

**Video 1.** Cell dividing parallel to the AP axis of the spinal cord. Confocal images through a mitotic cell in a 14-days regenerating axolotl spinal cord. DNA is labeled with Hoechst and shown as inverted grayscale. 1 µm z-steps. Images are displayed at 4 fps. Scale bar, 20 µm.

how this pathway may be contributing to the efficient, tube-like extension of the neural stem cell pool during regeneration. We first documented the spatiotemporal distribution of *Wnt5a* and *Wnt5b* transcripts by in situ hybridization. Intriguingly, we found graded upregulation of *Wnt5a* and *Wnt5b* transcripts precisely in the regeneration source zone 1 day after amputation (*Figure 3B*). The upregulation of *Prickle1* followed from day 3 and continued to be expressed in the caudal-most region of the regenerating spinal cord. We then endeavored to measure the size of the Prickle1 zone at different time-points of regeneration. Since it is not possible to capture the entire spinal cord length in sagittal sections of later stage regenerates, we determined the length of the Prickle1 positive zone by counting the number of consecutive cross-sections in which *Prickle1* in situ signal was visibly stronger in spinal cord cells than elsewhere (*Figure 3C*). As the regenerating spinal cord grew along its AP axis, the Prickle1 zone increased in size (429 ± 32 µm at day 3 and 981 ± 74 µm at day 6). At day 14, the Prickle1 zone covered the caudal-most ~1600 µm of the regenerating spinal cord (1579 ± 75 µm) but *Prickle1* transcripts returned to undetectable levels close to the amputation plane (*Figure 3C,D*). Remarkably, the Prickle1 zone correlated with regions of a simple neuroepithelial structure that have yet to initiate robust neurogenesis (*Figure 3D*).

## PCP is required for oriented cell divisions in the regenerating spinal cord

Next, we asked whether PCP-mediated mechanisms operate during regeneration. Because oriented cell divisions are a classical read-out of PCP in proliferating and elongating tissues (*Fischer et al., 2006*; *Gong et al., 2004*; *Matsuyama et al., 2009*; *Quesada-Hernandez et al., 2010*; *Saburi et al., 2008*), we set out to investigate the spatial aspects of cell division in the regenerating spinal cord. To measure cell division orientation, we produced image stacks across thick spinal cord cross-sections using confocal microscopy (*Video 1*). From these stacks we extracted two vectors, the vector between the two mitotic spindle poles of late anaphase or telophase cells (*S*) and the vector of the AP axis of the spinal cord (*AP*) (*Figure 4A*). From these vectors, we calculated the projection of the spindle on the AP axis ($Proj^{S}_{AP}$) (*Figure 4B* and see Materials and methods). This parameter ranges from 0, for cells dividing orthogonal to the AP axis, to 1, for cells dividing parallel the AP axis (*Figure 4C*). The $Proj^{S}_{AP}$ value was plotted for each mitotic cell according to its AP position along the spinal cord, what allowed us to record whether cells fell within or outside the Prickle1 zone (as measured in *Figure 3D*). We found that cells within the Prickle1 zone were significantly biased to divide parallel to the AP axis, whereas cells outside of the Prickle1 zone divided randomly (*Figure 4D*). These observations raised the possibility that PCP mechanisms orient the axis of cell division along the AP axis.

PCP mechanisms involve the polarized localization of the core PCP machinery. The function of PCP in many tissues has been tested via overexpression of core PCP components which presumably overwhelms the localization machinery and therefore disorients PCP (reviewed in [*Gray et al., 2011*]) (*Carreira-Barbosa et al., 2003*; *Cho et al., 2015*; *Goto and Keller, 2002*; *Gubb et al., 1999*; *Jessen et al., 2002*; *Tree et al., 2002*; *Wallingford and Harland, 2001*). Studies in *Drosophila* have classified PCP phenotypes into two classes. Deletion or overexpression of certain core PCP components such as Prickle randomize cell polarization cell autonomously only in those cells in which Prickle expression levels are deranged (*Adler et al., 2000*; *Gubb and Garcia-Bellido, 1982*; *Lawrence et al., 2004*; *Strutt and Strutt, 2007*). In contrast, a second class, which includes Vangl2,

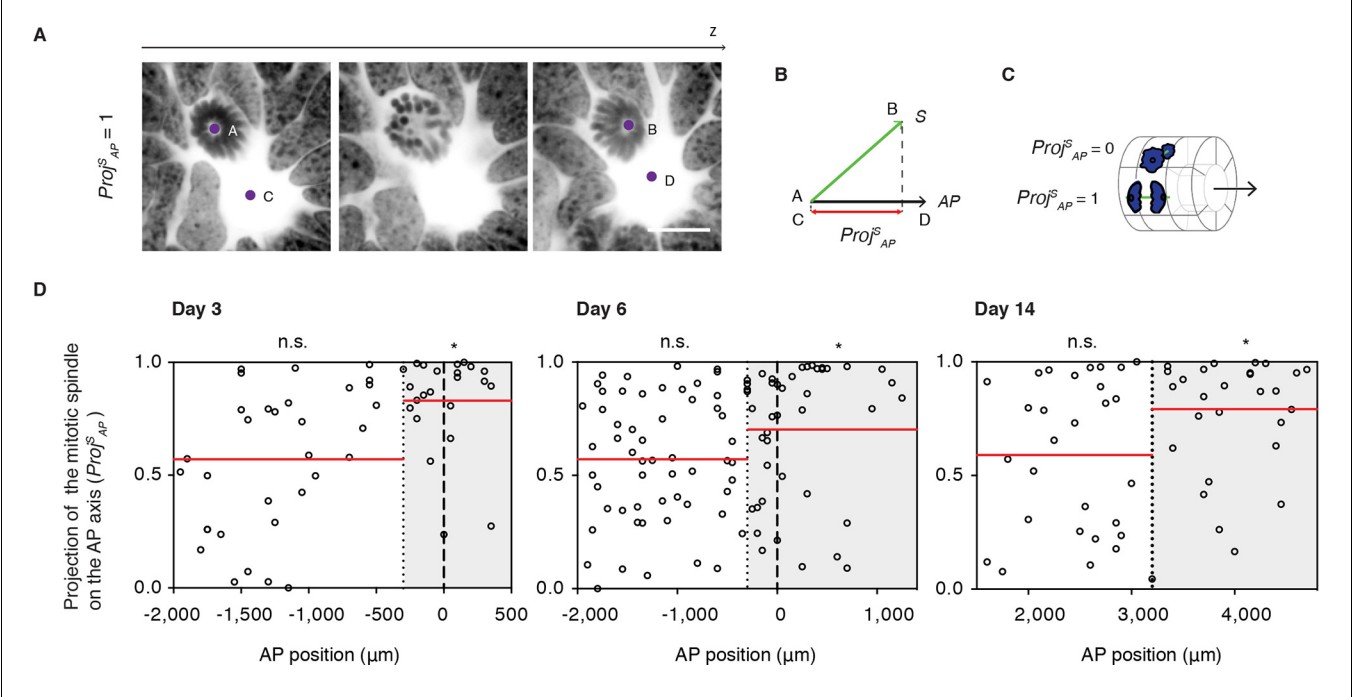

**Figure 4.** Cell divisions orient along the AP axis in the Prickle1 zone of the regenerating spinal cord. (**A**) Images show single optical confocal sections through a cell dividing along the AP axis regenerating axolotl spinal cord (from **Video 1**). DNA is labeled with Hoechst (inverted grayscale). Scale bar, 10 µm (**A,B**) The spindle vector (S) is defined by two points (A and B) obtained from the spindle poles, and the AP axis vector (AP) is defined by two points (C and D) in the center of the spinal cord central canal in the first and last optical slice of the z-stack. The projection of S on the AP vector ($Proj^{S}_{AP}$) was used as a proxy of cell division orientation. (**C**) $Proj^{S}_{AP}$ ranges from a value of 0, for divisions orthogonal to the AP axis, to 1, for divisions parallel to the AP axis. (**D**) Distribution of the projection of mitotic spindles on the AP axis at day 3, day 6, and day 14 after amputation. Each dot represents a cell division. Dotted lines mark the boundaries of the Prickle1 zone (gray boxes) as calculated in **Figure 3C**. Dashed lines mark the amputation planes. Red lines mark the mean value of the $Proj^{S}_{AP}$ along the AP axis within the Prickle1 zone and the Prickle1-negative zone. Note that cells divide preferentially oriented along the AP axis within the Prickle1 zone of the regenerating spinal cord (day 3 n=23 cells, day 6 n=43 cells, day 14 n=27 cells) but not in the Prickle-negative zone (day 3 n=32 cells, day 6 n=53 cells, day 14 n=29 cells). Cells pooled from at least 4 tails/group. Statistics within the Prickle1 zone and the Prickle1-negative zone to test whether the distribution deviates from uniform more than two standard deviations. * $p<0.05$, n.s. is not significant.

yields non cell-autonomous defects in PCP both in the cells with expression defects as well as their surrounding neighbors (**Vinson and Adler, 1987**). Similarly, the lack or excess of Vangl2 in vertebrates propagates polarity defects non-cell autonomously, spreading the PCP defect through the tissue (**Darken et al., 2002**; **Devenport and Fuchs, 2008**; **Mitchell et al., 2009**; **Montcouquiol et al., 2006**; **Sienknecht et al., 2011**). We took advantage of these two modes of disrupting PCP to study the cellular and tissue roles of PCP in the regenerating neural stem cells. To test a functional link between PCP and oriented cell division, we overexpressed *Prickle1* in spinal cord cells by electroporation of an axolotl *Prickle1* plasmid (**Figure 5A,B** and **Videos 2** and **3**). Quantitative real time PCR confirmed 10-fold higher *Prickle1* expression at day 6 in *Prickle1*-electroporated spinal cords than in controls (**Figure 5—figure supplement 1A**). However, because on average only 21.5% of the spinal cord cells overexpressed *Prickle1* (**Figure 5—figure supplement 1B**), the levels of *Prickle1* per electroporated cell were actually much higher. We focused on these electroporated, *Prickle1*-overexpressing cells to investigate whether excess *Prickle1* perturbed polarized cell divisions in the electroporated cells. Analysis of mCherry-only electroporated cells indicated that these cells displayed the same bias to divide parallel to the AP axis within the Prickle1 zone as in non-electroporated animals (**Figure 5C** and **Figure 4D**). In contrast, this bias in division orientation was lost in *Prickle1*-overexpressing cells (**Figure 5D**). *Prickle1*-overexpressing cells, however, continued to divide at a similar rate than control-electroporated cells (**Figure 5—figure supplement 2**). Thus,

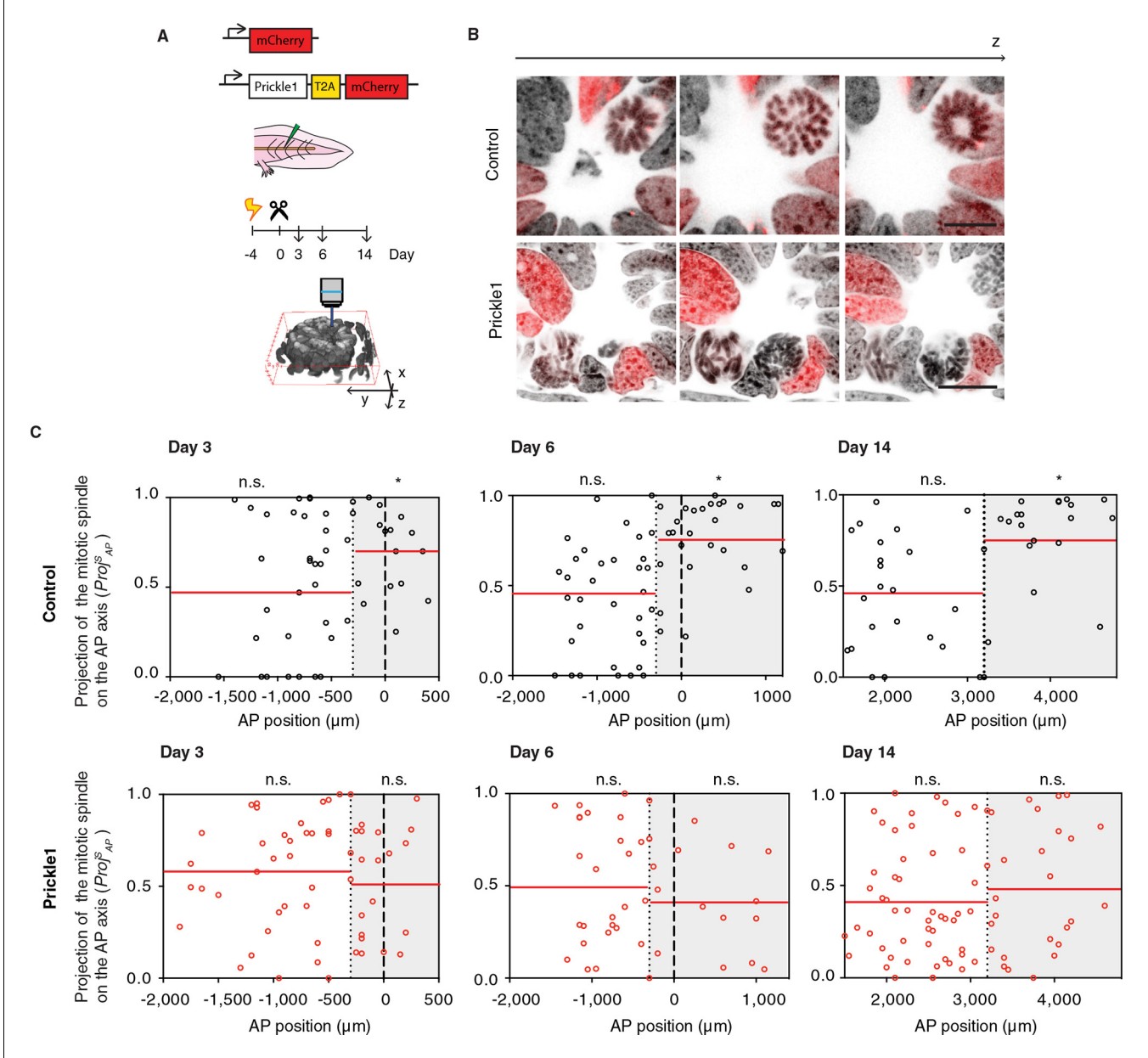

**Figure 5.** *Prickle1* overexpression randomizes cell division orientation in the regenerating spinal cord. (**A**) Spinal cord cells were electroporated either with a construct encoding for axolotl *Prickle1*, the T2A self-cleavage peptide, and mCherry, or mCherry alone. Amputated tails were allowed to regenerate 3, 6, and 14 days. Confocal z-stacks of electroporated cells in late anaphase or telophase were acquired for analysis. (**B**) Images show single optical confocal sections through representative dividing control cell (top, and *Video 2*) and *Prickle1*-overexpressing cell (bottom, and *Video 3*). Electroporated cells express nuclear mCherry (red). DNA is labeled with Hoechst (inverted grayscale). Scale bar, 10 μm. (**C,D**) Distribution of the projection of mitotic spindles on the AP axis in control (**C** and *Prickle1*-overexpressing cells (**D**) at day 3, day 6, and day 14 after amputation. Each dot represents a cell division. Dotted lines mark the boundaries of the Prickle1 zone (gray boxes) as calculated in *Figure 3C*. Dashed lines mark amputation planes. Red lines mark the mean value of $Proj^S_{AP}$. Note that control cells divide randomly within the Prickle1-negative zone (day 3 n=22 cells, day 6 n=34, day 14 n=22 cells), but show significant bias towards the AP axis within the Prickle1 zone (day 3 n=17 cells, day 6 n=27, day 14 n=21 cells). The bias to divide along the AP axis is lost in cells overexpressing *Prickle1* within the Prickle1 zone (day 3 n=22 cells, day 6 n=18 cells, day 14=28 cells). Cells pooled from at least 4 tails/group. Statistics within the Prickle1 zone and the Prickle1-negative zone to test whether the distribution deviates from uniform more than two standard deviations. * $p < 0.05$, n.s. is not significant.

The following figure supplements are available for figure 5:

**Figure supplement 1.** Characterization of Prickle1 expression in control and *Prickle1*-overexpressing spinal cords.

*Figure 5 continued on next page*

*Figure 5 continued*

**Figure supplement 2.** Prickle1 overexpression does not alter the cell cycle kinetics of regenerating neural stem cells.

PCP operates during spinal cord regeneration and one of its roles is to orient the axis of cell division along the rapidly elongating AP axis.

## PCP is essential for efficient elongation of the regenerating spinal cord

To examine the effect of interfering with PCP on overall spinal cord regeneration, we turned to over-expression of the core PCP component Vangl2, which is expected to disrupt the polarity of *Vangl2*-overexpressing cells as well as their neighbors, thus spreading the disruption of PCP to a larger proportion of spinal cord cells compared to the *Prickle1*-electroporated tails (*Darken et al., 2002*; *Mitchell et al., 2009*; *Montcouquiol et al., 2006*; *Sienknecht et al., 2011*). We co-electroporated *Vangl2* plasmid together with mCherry and mCherry alone as control, amputated the tails, and followed the regenerative growth of the spinal cord (*Figure 6A*). Measurements of spinal cord outgrowth at day 4 and day 6 showed that spinal cords overexpressing *Vangl2* were shorter than controls (*Figure 6B*). Since PCP instructs cells in the regenerating spinal cord to divide along the elongating AP axis, it is possible that disrupting PCP misplaces daughter cells and leads to less efficient AP extension. If that was the case, we expected to find more cells within the cross-sectional circumference of *Vangl2*-overexpressing spinal cords. Indeed, we counted a greater number of SOX2[+] cells on cross-sections of *Vangl2*-overexpressing spinal cords compared to controls (*Figure 6C*). Together, these observations point to the classical shorter and broader PCP phenotype and confirm that PCP mechanisms are required for efficient extension of the regenerating spinal cord.

## PCP is essential for maintaining the expansion of neural stem cells in the regenerating spinal cord

The orientation of the mitotic spindle sets the cleavage plane. The observation that the spindles oriented parallel to the AP axis implied that those spindles also tended to lie parallel to the apical surface of the spinal cord and therefore would have vertical cleavage planes. During development, such vertical cleavage planes with respect to the apical surface of the neural tube result in symmetric proliferative divisions, leading to the expansion

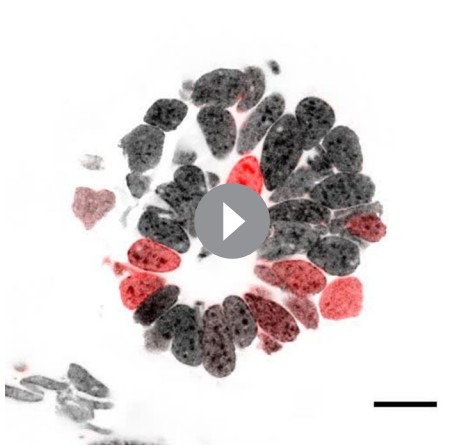

**Video 2.** Control cell dividing parallel to the AP axis of the spinal cord. Confocal images through a mitotic cell expressing nuclear mCherry (red) in the regenerating spinal cord, 14 days after amputation. DNA is labeled with Hoechst and shown as inverted grayscale. 1 μm z-steps. Images are displayed at 4 fps. Scale bar, 20 μm.

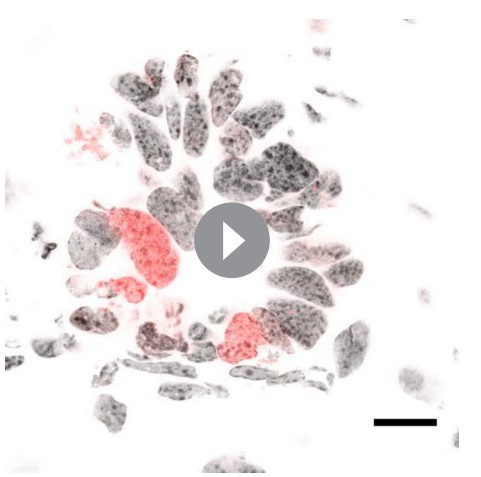

**Video 3.** *Prickle1*-overexpressing cell dividing orthogonal to the AP axis of the spinal cord. Confocal images through a mitotic cell electroporated with a construct encoding for axolotl *Prickle1*, the self-cleavage peptide T2A, and nuclear mCherry (red) in the regenerating spinal cord, 14 days after amputation. DNA is labeled with Hoechst and shown as inverted grayscale. 1 μm z-steps. Images are displayed at 4 fps. Scale bar, 20 μm.

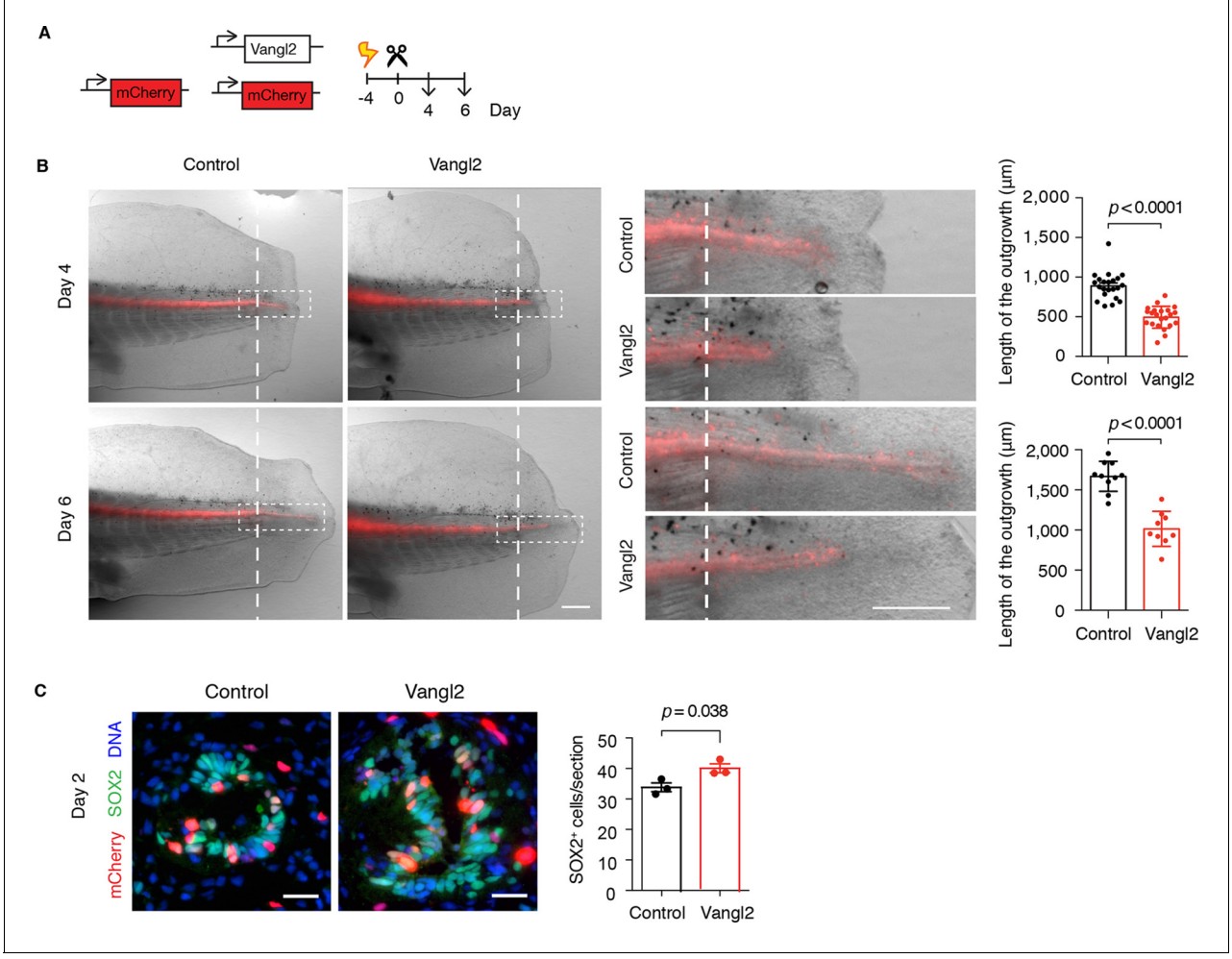

**Figure 6.** *Vangl2* overexpression yields shorter and broader spinal cord outgrowth. (**A**) Axolotl spinal cords were co-electroporated with *Vangl2* and mCherry or mCherry alone as control, tails were amputated, and monitored during regeneration. (**B**) Representative images of control and *Vangl2*-overexpressing spinal cords at day 4 and day 6. Insets show spinal cord outgrowths. Dashed lines mark the amputation planes. Mean ± s.e.m. of spinal cord length on the right. Dots represent individual axolotls, pooled from two independent experiments. *P* values indicate the result of two-tailed unpaired Student's *t*-tests. Scale bars, 500 µm. (**C**) Immunofluorescence on cross-sections across the source zone of control and *Vangl2*-overexpressing spinal cords 2 days after amputation. SOX2 labels neural stem cells, mCherry labels electroporated cells, and Hoechst labels DNA. Mean ± s.e.m. of the number of SOX2$^+$ cells per cross-section on the right. N=3 tails/group. *P* value indicates the result of two-tailed unpaired Student's *t*-test. Scale bars, 20 µm.

of the early neuroepithelial cell pool (*Chenn and McConnell, 1995*; *Das and Storey, 2012*; *Peyre and Morin, 2012*; *Xie et al., 2013*). We thus asked whether the shortened spinal cord in *Vangl2*-overexpressing axolotls was due solely to reduced morphogenetic extension along the AP axis, or whether randomizing the axis of cell division additionally caused defects related to stem cell self-renewal and expansion. In order to confirm disruption of the orderly spindle orientation by Vangl2 overexpression, we first examined the cleavage plane of mitotic cells within the source zone of control and *Vangl2*-overexpressing spinal cords 4 days after tail amputation (*Figure 7A,B*). We found a tight distribution of vertically oriented cleavage planes in control-electroporated spinal cords, whereas mitotic cells within the regenerating stem cell pool of *Vangl2*-overexpressing spinal cords displayed a more variable distribution of cleavage planes (*Figure 7C*), indicating a relaxation in the constraint to divide with vertical cleavage planes.

To investigate whether randomizing cleavage planes led to proliferation defects, we first quantified the number of phospho-histone H3 (PH3) cells in control and *Vangl2*-overexpressing spinal cords in different regions along the regenerating spinal cord (*Figure 7D*). Interestingly, at day 4 the

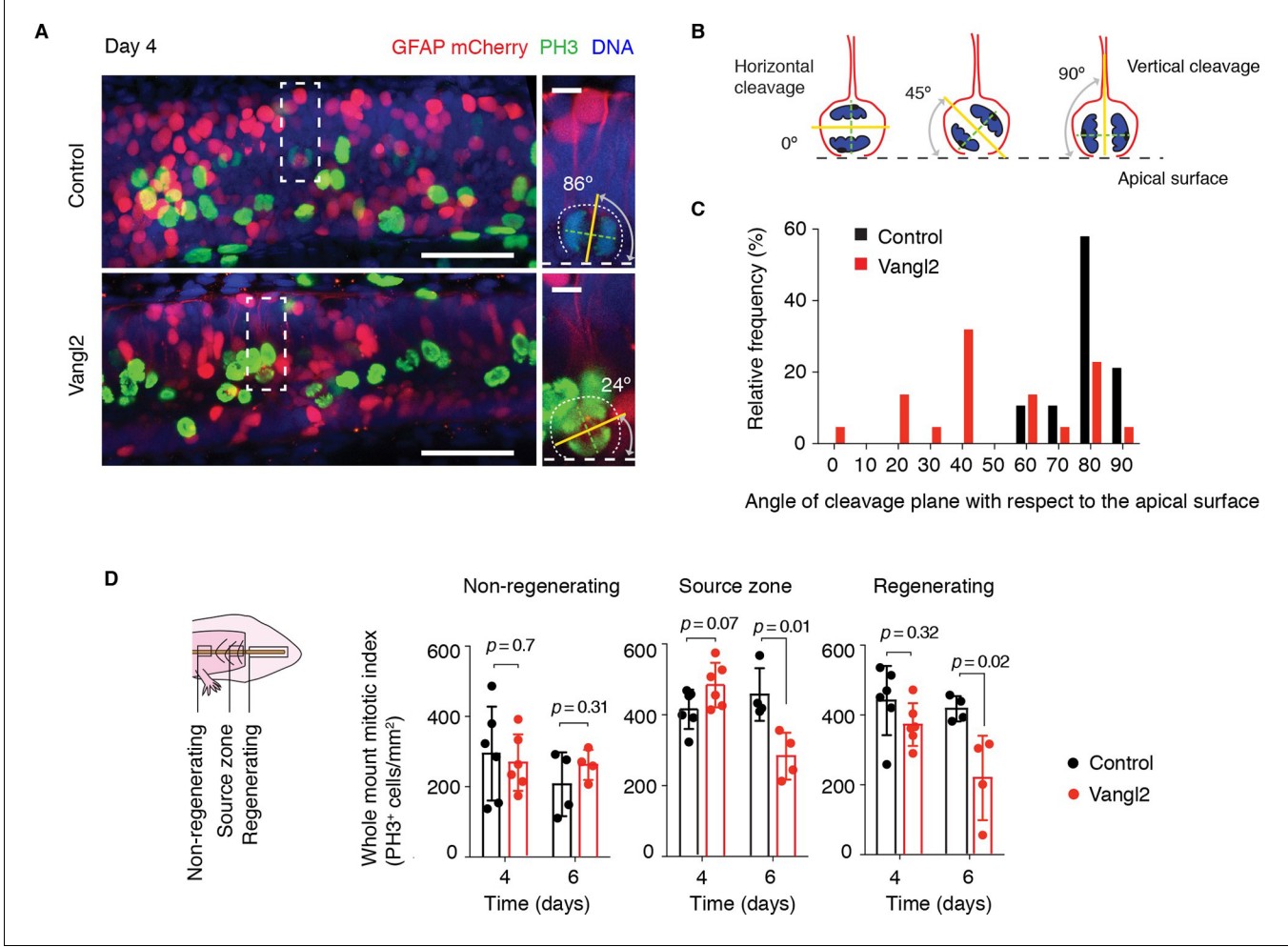

**Figure 7.** *Vangl2* overexpression randomizes cleavage planes and halts the expansion of the regenerating neural stem cell pool. (A) Longitudinal optical sections of the source zone from control and *Vangl2*-overexpressing spinal cords at day 4 after amputation. PH3 labels late G2 and mitotic cells in green, GFAP labels the basolateral membrane of neural stem cells in red, nuclear mCherry labels electroporated cells in red, Hoechst labels DNA in blue. Insets show representative PH3+mitotic cells within each group. Dashed white lines mark the apical surface of the spinal cord. Dotted lines outline cells. Scale bars, 100 µm. Insets, 10 µm. (A,B) The mitotic spindle (green dashed lines) was deduced from late anaphase or telophase cells (see Materials and methods). A vector orthogonal to the mitotic spindle was drawn as cleavage plane (yellow lines). The angle of cleavage plane was determined as the angle between the cleavage plane and apical surface of the spinal cord. (C) Distribution of cleavage plane angles within the source zone in control (n=19 cells from 4 tails) and *Vangl2*-electroporated spinal cords (n=22 cells from 4 tails). The distribution of cleavage angles in *Vangl2*-electroporated spinal cords is more variable and significantly differs from the tighter distribution of control cells (*p*<0.0001, Mann-Whitney U test). (D) Quantification of PH3+ cells in non-regenerating and regenerating regions of control and *Vangl2*-electroporated spinal cords, 4 and 6 days after amputation (see Materials and methods). Dots denote individual tails. Data, mean ± s.e.m of n=6 tails at day 4 and n=4 tails at day 6 per group. *P* values indicate the result of two-tailed unpaired Student's *t*-tests (Vangl2 versus control, for each region).

The following figure supplement is available for figure 7:

**Figure supplement 1.** Quantification of apoptotic cells in control and *Vangl2*-electroporated spinal cords during regeneration.

density of PH3+ cells was similar in control and *Vangl2*-overexpressing spinal cords. By day 6, however, the density of PH3+ cells was significantly reduced in the regenerative regions of *Vangl2*-overexpressing spinal cords compared to controls, that must undergo symmetric proliferative divisions. This reduction was not due to increased apoptosis as we did not observe a significant difference in the number of caspase-3+ cells between control and Vangl2-overexpressing spinal cords up to this time point (*Figure 7—figure supplement 1*). These data suggested that disruption of PCP did not affect the onset of rapid proliferative divisions, since in both samples, we observed the initial

increase in mitotic figures at day 4 after amputation. The later appearance of a lower frequency of mitoses in the *Vangl2*-overexpressing in day 6 regenerates suggested an effect unrelated to the initial acceleration of the cell cycle.

These observations raised the possibility that altering the axis of cell division by overexpressing *Vangl2* resulted in premature neurogenic divisions that would gradually reduce the number of proliferative cells during regeneration. To label proliferating cells and their progeny, we injected *Vangl2* and control electroporated axolotls with the thymidine analog ethynildeoxyuridine (EdU) daily from day 3 to day 8 and collected the tissue 1 hour later for analysis (*Figure 8A*). To identify newborn neurons we immunostained spinal cord cross-sections for the early neuronal marker Tuj1 (*Figure 8B*). In agreement with BrdU experiments (*Figure 2H*), a substantial number of EdU$^+$ cells differentiated into neurons within the 6-day labeling period in the non-regenerating region of the spinal cord, and *Vangl2* overexpression had no effect on the fraction of neurons formed (*Figure 8B*). However, while practically no neurons were born during the labeling time in the regenerating region of control spinal cords, we found a significant number of newborn neurons in *Vangl2*-overexpressing spinal cords (*Figure 8B,C*). Taken together, these findings indicate that PCP is an essential mediator of the switch from the neurogenic divisions seen in the uninjured spinal cord to the proliferative divisions observed in the regenerating spinal cord at least during the first 8 days of regeneration. Perturbing PCP by overexpression of *Vangl2* results in more variable mitotic cleavage planes and premature neurogenesis that presumably reduces the numbers of proliferative cells and hampers regenerative spinal cord growth.

## Discussion

While spinal cord neural stem cells in regenerative as well as non-regenerative vertebrates all respond to spinal cord injuries, the ultimate outcome differs markedly and only the axolotl faithfully restores entire regions of missing spinal cord (*Becker and Becker, 2015*; *Tanaka and Ferretti, 2009*). This overall difference in regenerative success among vertebrates is largely reflected in differences in daughter cell fate decisions. In mammals, spinal cord lesions induce neural stem cells to proliferate and give rise mainly to scar-forming astrocytes and oligodendrocytes (*Barnabe-Heider et al., 2010*; *Meletis et al., 2008*). The resulting glial scar tissue, although important for maintaining the integrity of the injured spinal cord (*Sabelstrom et al., 2013*), inhibits axon regrowth limiting functional recovery (*Burda and Sofroniew, 2014*; *Frisen et al., 1995*). In zebrafish, neural stem cells react to spinal cord lesions by increasing their local neurogenic output (*Kuscha et al., 2012*; *Reimer et al., 2013*; *Reimer et al., 2008*). Although a thin glial 'bridge' restores tissue integrity and function (*Goldshmit et al., 2012*), the spinal cord tissue architecture is not restored after lesion. An important question is which stem cell properties and what molecular pathways are important for faithful spinal cord regeneration.

Here, we find that complete restoration of missing regions of the axolotl spinal cord involves the re-expression of developmental cassettes associated with embryonic neuroepithelial cells of the PNT (*Figure 1*). The striking correspondence between downregulated genes at the onset of regeneration with those that were low in the embryonic SZ+PNT (*Olivera-Martinez et al., 2014*) suggests that regenerating neural stem cells lose pro-neurogenic features and acquire a more multipotent phenotype. Indeed, clonal tracing studies (*McHedlishvili et al., 2007*) and the fact that implanted clonally-derived neurospheres can reconstitute a complete spinal cord after tail amputation previously pointed out that multipotent neural stem cells exist during regeneration in the axolotl (*McHedlishvili et al., 2012*).

The correspondence between upregulated genes was significant but lower than for the downregulated genes. It should however be noted that the SZ+PNT chick dataset included genes associated with neuromesodermal progenitors, that during the formation of the vertebrate body axis generate neural and paraxial mesoderm tissues (*Wilson et al., 2009*). Transcripts of genes characteristic of this dual-fated cell population such as the mesodermal factors *T/Bra* (*Martin and Kimelman, 2010*) and *Msgn1* (*Chalamalasetty et al., 2014*) were absent or scored very low counts in our dataset (see *Figure 1—source data 1*), suggesting that neural stem cells in the axolotl spinal cord either do not transit through a neuromesodermal cell state or that such population is very small. Taking this into consideration, the actual correspondence between the PNT state and the axolotl regenerating neuroepithelial state is likely underestimated in our comparison.

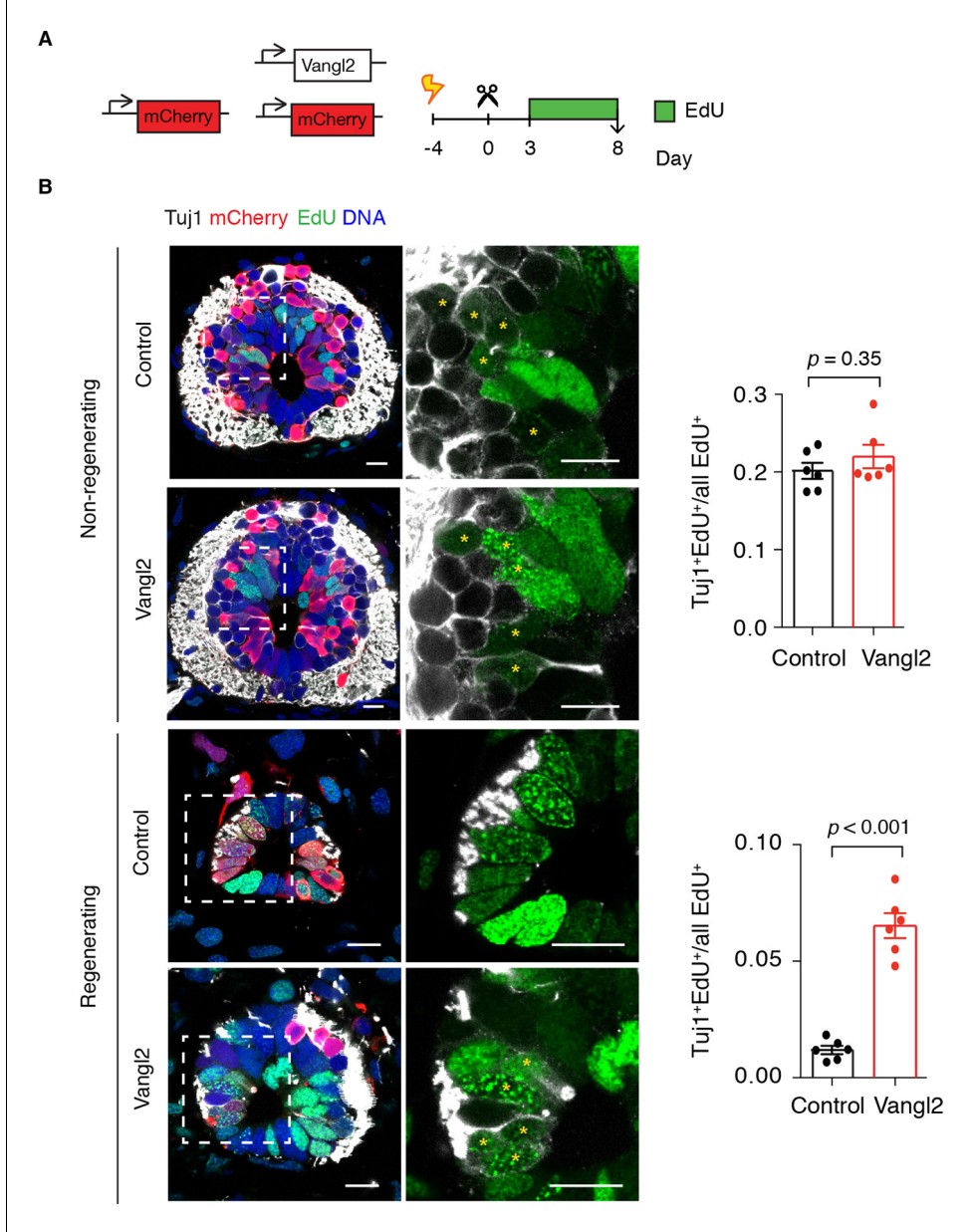

**Figure 8.** *Vangl2* overexpression leads to premature neurogenesis in the regenerating spinal cord. (**A**) Axolotls with mCherry or mCherry and *Vangl2* electroporated spinal cords were injected with EdU daily, from day 3 to 8, to label proliferating cells and trace their progeny. (**B**) Representative images of non-regenerating and regenerating regions of control and *Vangl2*-overexpressing spinal cords. Tuj1 (in white) labels early-born neurons, EdU (green) labels cells that underwent DNA synthesis and their progeny, mCherry (red) labels electroporated cells and Hoechst (blue) labels DNA. Asterisks mark neurons generated during the EdU labeling time. Quantification of the fraction of EdU$^+$ cells that differentiated into neurons (Tuj1$^+$EdU$^+$) over total EdU$^+$ cells is shown on the right. Note that in non-regenerating regions there is no significant difference in the fraction of EdU$^+$ cells that differentiate into neurons between control and *Vangl2*-overexpressing spinal cords (0.20 ± 0.01 and 0.22 ± 0.02 respectively, $P$=0.345). However, the overexpression of *Vangl2* leads to premature neurogenesis in the regenerating portion of the spinal cord compared to controls (0.07 ± 0.01 in Vangl2 versus 0.01 ± 0.01 in controls, $P$<0.001). Data, mean ± s.e.m. of n=6 tails, pooled from two independent experiments. Statistics, two-tailed unpaired Student's *t*-test (Vangl2 versus control, for each region). Scale bars, 20 μm.

Although regeneration-specific factors may exist, our study shows that the redeployment or fine-tuning of signals that played a role during embryonic development are required for spinal cord regeneration in the axolotl. For example, during development the onset of retinoic acid signalling halts the expansion of neural progenitors and drives them to make neurons in the developing spinal cord. In our gene expression analysis, retinoic acid-target genes such as *Rarb, Crabp1,* and *Meis2,* become downregulated in the regenerating spinal cord, suggesting that signals that promote neurogenesis programs are shut down to possibly allow the execution of self-renewing, multipotent programs during regeneration. In the embryo, the somites produce retinoic acid. In the axolotl, the tail muscles might continue supplying retinoic acid postnatally. The decline in retinoic acid signaling that we observe during regeneration may be related to the fragmentation of the flanking muscle fibers after amputating the tail, that degenerate and might therefore stop supplying retinoic acid. This would allow the spinal cord cells close to the amputation site to re-acquire a multipotent state and grow into the tail blastema. Intriguingly, we noticed that when muscles start regenerating at the amputation site, neural stem cells start resuming neurogenesis. Further studies will be needed to address the relationship between mesodermal tissue, neurogenesis, and retinoic acid.

Besides the loss of pro-neurogenic traits and signals, instructive factors must orchestrate and organize the newly forming tissue during regeneration as they do during embryonic development. An important aspect of our findings is that, during regeneration, the implementation of PCP mechanisms coordinately orients tubular outgrowth and switches the mode of cell division from a neurogenic to an expansive, proliferative mode of division. PCP has previously been implicated in the extension of tubular structures such as kidney tubules (*Fischer et al., 2006*; *Karner et al., 2009*; *Lienkamp et al., 2012*) and on the other hand has been associated with expansion of muscle satellite stem cells (*Le Grand et al., 2009*). In the developing mouse brain, Vangl2 malfunction interferes with normal spindle positioning and leads to premature terminal differentiation of neural progenitors (*Lake and Sokol, 2009*). In the regenerating axolotl spinal cord both aspects of PCP appear to operate simultaneously to elicit efficient expansion of the neural stem cell pool in a tubular form. Although here we have focused on the role of PCP in controlling the orientation of cell division, the extension of the regenerating spinal cord tube may also rely on other proliferation-independent, PCP-oriented cell behaviors such as convergence and extension (*Wallingford, 2012*).

In the vertebrate embryo, Wnt5a gradients often establish PCP to orient cell behaviors driving the elongation of multiple developing tissues (*Gao et al., 2011*; *Yamaguchi et al., 1999*). Intriguingly, graded Wnt5a and Wnt5b upregulation is a common feature of regenerative systems such as the lizard tail (*Hutchins et al., 2014*), the salamander limb (*Ghosh et al., 2008*; *Knapp et al., 2013*; *Wu et al., 2013*), and the tail and spinal cord of salamander, fish, and *Xenopus* (this study and *Caubit et al., 1997*; *Stoick-Cooper, 2007*; *Hui et al., 2014*; *Sugiura, 2009*). Yet, the role of Wnt5 in the regeneration field has remained controversial. In zebrafish fin regeneration, constitutive overexpression of Wnt5 halts regeneration while in *Xenopus*, grafting Wnt5-expressing cells yields the generation of a complete ectopic tail (*Stoick-Cooper et al., 2007*; *Sugiura et al., 2009*). Our study suggests that Wnt5 gradients may implement PCP to re-establish the axis of polarity needed to drive polarized, regenerative tissue outgrowth. It is thus possible that the contradictory results in *Xenopus* and zebrafish studies are due to the experimental approach taken to investigate the role of Wnt5a. In the fin, global overexpression of Wnt5a inhibited regeneration, which was interpreted to mean that non-canonical Wnt signalling inhibits regeneration. However, since as we and others have shown, excess of PCP signaling has negative effects on regeneration and so it is likely that this global overexpression was in fact deranging PCP, leading to inefficient regeneration. Alternatively, the relative wiring between the canonical and non-canonical branches of the Wnt signalling pathway may be different in the zebrafish fin and spinal cord compared to the regenerating amphibian tail leading to different influences on regeneration.

Here, by examining regenerative behaviors at cellular rather than tissue scale, we have found that the acquisition of an embryonic PNT-like signature includes the implementation of PCP mechanisms that set the orientation and self-renewal properties of regenerative neural stem cells in the axolotl. In sum, our work points to concepts and molecular pathways that may be used in the future to divert endogenous neural stem cells in other vertebrates toward a regenerative phenotype aimed at restoring the damaged nervous system.

## Materials and methods

### Axolotls

Axolotls, *Ambystoma mexicanum*, 3 cm in length snout-to-tail were used for experiments. Axolotls were kept in tap water in individual cups and fed daily with Artemia. Before any manipulation or imaging, axolotls were anaesthetized in 0.01% benzocaine. The axolotl animal work was performed under permission granted in animal license number DD24-9168.11-1/2012-13 conferred by the Animal Welfare Commission of the State of Saxony, Germany (Landesdirektion, Sachsen).

### Tissue microdissection and RNA extraction

The tails of uninjured, 1 day, and 6 days regenerating axolotls were collected and submerged in RNA*later* (QIAGEN) to protect their RNA. To generate day 0 samples, 500 μm of spinal cord at the level of the tail myotome 15 from the cloaca were microdissected from uninjured tails. To generate day 1 samples, the 500 μm of spinal cord anterior to the amputation plane, corresponding to the regeneration source zone, were microdissected from 1-day regenerating tails. To generate day 6 samples, the whole regenerated spinal cord was microdissected from 6-days regenerating tails. In every case, 48-50 pieces of spinal cord were pooled for each of three biological replicates. Total RNA was extracted using TRIzol (Life Technologies) and purified using RNeasy MinElute cleanup kit (QIAGEN). RNA quality was assessed using a bioanalyzer (Agilent Technologies) and only samples with a RNA integrity number (RIN) above 9.5 used for NanoString analysis.

### Gene set selection and NanoString probe design

From the list of genes that undergo major regulatory changes at the onset of neurogenesis in the developing chick spinal cord (*Olivera-Martinez et al., 2014*), we selected 100, the 50-most upregulated and the 50-most downregulated genes for which we found axolotl orthologs in our transcriptome assembly. An axolotl-specific custom nCounter CodeSet, including *Rpl4* as housekeeping gene, was designed by the NanoString bioinformatics team. See *Supplementary file 3* for the complete list of target sequences.

### NanoString nCounter assay

NanoString nCounter technology was chosen for gene expression profiling (*Geiss et al., 2008*). Briefly, 50 ng of total RNA per group (three biological replicates) was hybridized for 20 hours with the 100-gene custom nCounter CodeSet. Hybridized RNA transcripts were purified and processed in the nCounter Prep Station and quantified in the nCounter Digital Analyzer following NanoString's gene expression assay protocol (www.nanostring.com).

### NanoString data analysis

NanoString data needs to be normalized before performing differential expression analysis. To correct for technical variation in hybridization and purification efficiency, raw counts were first normalized using the geometric mean of the internal, spike-in positive control probes. Next, to determine the presence or absence of gene expression, a one-tailed Student's t-test of the mean counts of each gene against the mean counts of negative controls was performed. Counts for genes for which $p \geq 0.05$ were considered not significant and blanketed to 1. Counts for genes for which $p < 0.05$ were considered significant and the mean plus three standard deviations of the negative control probes was subtracted to remove the background noise inherent to the method. Count values that fell below 1 were blanketed to 1.

Last, counts were normalized to the housekeeping gene *Rpl4* to account for input material variation and set ready for differential gene expression analysis.

Differential expression analysis was carried out using the DESeq package (*Anders and Huber, 2010*) in R (http://www.r-project.org/). Count data were modeled by negative binomial distributions and negative binomial tests were used to determine statistical significance. A gene was considered significantly regulated when $p < 0.05$ (corresponding to a Benjamini-Hochberg adjusted $p < 0.07$) (*Benjamini and Hochberg, 1995*).

## RNA in situ hybridization and probes

Axolotl tails were fixed in fresh MEMFA at 4°C for 72 hours. 10 µm paraffin sections were used for in situ hybridizations as previously described (*Roensch et al., 2013*). RNA probes were synthesized from clones in our expressed sequence tag (EST) library harboring axolotl *Wnt5a* (GenBank accession number: Z14047.1), *Wnt5b* (Z14048.1), *Prickle1* (CO780858.1), *Pax6* (CO784109.1), *Pax7* (AY523019.1), *Gfap* (CO780148.1), *Egr1* (JK976899.1), *Cdx4* (CO778950.1), and *Greb1* (JK980018.1).

## Immunostaining

Tails were fixed in MEMFA overnight at 4°C, rinsed in PBS, cryopreserved in 30% sucrose in PBS at 4°C, and frozen in Tissue-Tek (O.C.T Compound, Sakura Finetek) for cryosectioning. Sections were collected on adhesive slides and dried at room temperature for several hours. Slides were then washed 3 times in PBS/0.3% Triton X-100, and incubated in glycine buffer for 20 minutes. After that, antigen retrieval was carried out boiling the sections in citrate buffer pH 6 (DakoCytomation, Dako) for 5 minutes. The sections were allowed to cool down and blocked in blocking buffer (PBS/0.3% Triton X-100/1% goat serum). After 1 hour, primary antibodies were added diluted in blocking buffer and incubated overnight at 4°C. Several PBS washes later, secondary antibodies were added diluted in blocking buffer and incubated for 2 hours at room temperature. Hoechst (1 mg/ml stock solution diluted 1:500) was added together with the secondary antibodies to stain nuclei. After 3 PBS washes, slides were mounted in 50% glycerol/PBS.

For whole-mount immunostaining, the skin and muscles were removed from one side of the tail to expose the spinal cord and facilitate antibody penetration. Washing steps were 1-1.5 hours each and antibody incubations 48 hours. The tails were mounted using plastic paraffin between the slide and coverslip to avoid squeezing the tissue.

Antibodies used were: rabbit anti-SOX2 (homemade as described in [*McHedlishvili, 2012*], 1:5000), mouse anti-GFAP (Chemicon #MAB360, 1:400), mouse anti-CLDN5 (Invitrogen #35-2500, 1:500), rabbit anti-OCLN (Invitrogen 71-1500, 1:500), mouse anti-ZO1 (BD Biosciences #610966, 1:200), rabbit anti-phospho histone H3 (Millipore #06-570, 1:400), mouse anti-PCNA (Santa Cruz Biotech sc-56, 1:400), mouse anti-Tuj1 (R&D Systems MAB1195, 1:500), rabbit anti-active Caspase 3 (Abcam, ab13847, 1:800).

## Mitotic index

Tails of uninjured axolotls and 3-, 4-, 6-, and 8-days regenerates were fixed and prepared for cryosectioning. Consecutive 50-µm cross-sections were collected from the ~2 mm of tissue anterior to myotome 15 from uninjured tails and from the tip of the regenerating spinal cord to the 500 µm source zone from regenerating tails. Immunostaining for SOX2 was used to label neural stem cells and PCNA to label proliferating cells. Hoechst was used to label nuclei and cells in M-phase were identified by chromatin condensation. Using confocal microscopy, every section was scanned and the number of SOX2$^+$PCNA$^+$ mitotic cells was annotated. Single optical sections were taken from non-consecutive sections to count the number of SOX2$^+$PCNA$^+$ cells per cross-section. Counts were carried out using Fiji (http://fiji.sc/Fiji) (*Schindelin et al., 2012*).

The length of individual cells in the AP dimension was determined for cells from the uninjured cell population and from regenerating and non-regenerating cell populations 3 days after amputation by counting the number of 1 µm optical sections that encompassed individual nuclei (3 biological replicates per group, 10-25 cells per replicate). As the mean cell length was approximately equal for all cell populations (see http://nbviewer.jupyter.org/gist/fabianrost84/3cc58a27b5688f4e2eba), the number of proliferating cells in a section $k$ from a biological replicate $i$ was estimated by:

$$N_P^{k,i} = N_{PCNA}^{k,i} \frac{l_s}{l_c},$$

where $N_{PCNA}^{k,i}$ is the number of proliferating cells in a cross-section $k$, $l_s$ is the thickness of the cross-section and $l_c$ is the mean length of a cell in the AP dimension. Then, the mitotic index in a section is given by

$$mi_{section}^{k,i} = \frac{N_M^{k,i}}{N_P^{k,i}},$$

where $N_M^{k,i}$ is the number of mitotic cells in the section $k$ from a biological replicate $i$. For a specific cell population under study, delimited by two positions along the AP axis $l_{min}$, and $l_{max}$, the mitotic index was determined for a number of $n_s$ sections. From those, the mean mitotic index ($mi_{replicate}^i$) was calculated. The 1-$\sigma$ error was estimated by the standard error of the mean with finite population size correction (*Edgeworth, 1918*):

$$\Delta mi_{replicate}^i = \sqrt{\frac{n_s^{total} - n_s}{n_s^{total} - 1}} \frac{\sigma^i}{\sqrt{n_s}},$$

where $\sigma^i$ is the standard deviation of the mitotic index of the $n_s$ sections and $n_s^{total}$ is the total number of sections in the specific cell population under study:

$$n_s^{total} = \frac{l_{max} - l_{min}}{l_s}.$$

The average mitotic index of a specific cell population at a given time (*mi*) was obtained as the mean mitotic index of the $n$ biological replicates at this time point and the corresponding 1-$\sigma$ error was extracted by summing the intra- and interexperimental error:

$$\Delta mi = \Delta mi_{mean}^i + \frac{\sigma}{\sqrt{n}}.$$

Where $\Delta mi_{mean}^i$ is the mean intraexperimental error and $\sigma$ is the standard deviation of the mitotic index of the $n$ biological replicates. The raw data and the calculations can be found in http://nbviewer.jupyter.org/gist/fabianrost84/3cc58a27b5688f4e2eba.

## Mitotic index from whole mounts

Stacks of confocal images of the spinal cord were acquired for counting PH3$^+$ cells. Counts representative of the non-regenerating cell population were obtained from a 500 µm bin located 1.5 mm anterior to the amputation plane. Counts representative of the source zone were obtained from a 500 µm bin anterior to the amputation plane. Counts for the regenerating cell population were obtained from confocal stacks of the regenerated spinal cord. The number of PH3$^+$ cells was counted from the maximum intensity projections of the image stacks and the mitotic index calculated as the number of PH3$^+$ cells per area. Image processing, cell counts, and area measurements were carried out in Fiji.

## BrdU labeling, immunostaining, and counting

BrdU (Sigma-Aldrich) was dissolved at 2.5 mg/ml in PBS. A series of BrdU injections were given intraperitoneally to uninjured and regenerating axolotls, from day 6 after amputation. Adapting the labeling conditions defined by Nowakowski (*Nowakowski et al., 1989*), uninjured axolotls were injected at 24 hours intervals up to 220 hours and regenerating axolotls at 12 hours intervals up to 120 hours. The tail tissue of at least 5 axolotls was harvested at the indicated time-points, fixed, and cryosectioned in 10 µm cross-sections. From uninjured tails, sections were collected at the level of the myotome 15. From regenerating tails, sections were collected from the newly regenerated spinal cord, as regenerating samples, and from at least 1.5 mm anterior to the amputation plane, as non-regenerating samples.

After several PBS washes and quenching with 0.1 M glycine-Tris (pH 7.4), sections were treated with 2M HCl 15 min at 37°C to denature DNA. BrdU was detected with a mouse anti-BrdU monoclonal antibody directly coupled to rhodamine (homemade), after overnight incubation at 4°C. Hoechst was used to label DNA.

Every third section was imaged with a Zeiss Axio Observer microscope using a Zeiss Plan-Apochromat 40x 0.95 objective. The fraction of BrdU-labeled cells was calculated as the number of BrdU-labeled cells surrounding the central canal of the spinal cord. Rounded nuclei, characteristic of neurons, were excluded from the analysis. Counts were carried out in Fiji.

## Cell cycle length and cell cycle parameters analysis

The modeling framework developed by Lefevre and co-workers (*Lefevre et al., 2013*) which is, in turn, an adaptation of the model by Nowakowski and colleagues (*Nowakowski et al., 1989*) was extended to extract the cell cycle parameters from the cumulative BrdU labeling data.

Under the assumptions that (i) the length of the cell cycle and the cell cycle phases are constant for all cells, that (ii) divisions occur completely asynchronous and that (iii) all cells are proliferating the model by Lefevre et al. predicts the fraction of BrdU-labeled cells, $R'$:

$$R'(t) = \begin{cases} \dfrac{r^{\frac{T_{G2+M}+T_s}{T_c}} - r^{\frac{T_{G2+M}-t}{T_c}}}{r-1} & 0 \leq t <=< T_{G2+M} \\[2em] 1 - \dfrac{r^{\frac{T_c+T_{G2+M}-t}{T_c}} - r^{\frac{T_{G2+M}+T_s}{T_c}}}{r-1} & T_{G2+M} \leq t < T_c - T_s \\[1em] 1 & t \geq T_c - T_s \end{cases},$$

if $r \neq 1$ and

$$R'(t) = \begin{cases} \dfrac{T_s+t}{T_C} & 0 \leq t < T_c - T_s \\[1em] 1 & t \geq T_c - T_s \end{cases},$$

if $r = 1$, where $t$ is the time after the start of BrdU injections, $T_C$ is the cell cycle length, $T_S$ is the length of S-Phase, $T_{G2+M}$ is the combined length of G2 and M-phase and $r$ is the average number of daughter cells remaining proliferating progenitor cells.

We assumed asymmetric neurogenic cell divisions, *i.e.*, each cell division produces only one proliferating cell and one non-proliferating cell, for the uninjured and non-regenerating cell populations and proliferative cell divisions, *i.e.*, each cell division produces two proliferating cells, for the cell population tissue. Hence, $r = 1$ for the uninjured and non-regenerating cell populations and $r = 2$ for the regenerating cell population.

As there are also non-proliferating cells, we have to correct the labeling $R$ with the growth fraction $g$:

$$R(t) = g(t)R'(t).$$

In the case of proliferative cell divisions the growth fraction will shrink over time because the proliferating cells increase and the non-proliferating cells remain constant in number. Hence,

$$g(t) = \frac{GF r^{\frac{t}{T_c}}}{GF r^{\frac{t}{T_c}} + 1 - GF},$$

where GF is the initial growth fraction at $t = 0$. In the case of neurogenic divisions the growth fraction will be constant, hence $g(t) = GF$.

We estimated the parameters for each cell population by the method of least squares, that is, minimizing the sum of squared errors (*SSE*) calculated as follows:

$$SSE(T_s, T_c, T_{G2+M}, GF) = \sum_{i=1}^{m} \left( \frac{Rexp_i - R(t_i|T_s, T_c, T_{G2+M}, GF)}{\sigma_i} \right)^2,$$

where $m$ is the number of measurements, $Rexp_i$ is the measured BrdU-labelled fraction of the $i$-th measurement at time $t_i$ and $\sigma_i$ is the standard deviation of the measurements at this time point. We minimized the *SSE* using the function migrad from the python module iminuit (https://github.com/iminuit/iminuit). This also allowed us to determine the 68% confidence intervals for the fitted parameters based on the parabolic approximation. The BrdU labeling data and the implementation of the parameter estimation can be found in http://nbviewer.jupyter.org/gist/fabianrost84/3cc58a27b5688f4e2eba.

Because all mitotic cells were BrdU-labeled 9 hours after the start of the BrdU labeling we estimated the combined length of G2- and M-phase as $T_{G2+M}$ = 9 hours. We assumed that cells were homogenously distributed in the cell cycle. Hence, M-phase length is given by $T_M = mi \cdot T_c$, where $mi$ is the mitotic index and $T_c$ is the cell cycle length. The length of G1-phase can then be calculated as $T_{G1} = T_c - T_{G2+M} - T_s$. We estimated the errors of $T_M$ and $T_{G1}$ by linear error propagation (see http://nbviewer.jupyter.org/gist/fabianrost84/3cc58a27b5688f4e2eba).

## Plasmid constructs

*Prickle1* was found in our EST library (clone BL010D_C10) (*Habermann et al., 2004*). *Prickle1* open reading frame (ORF) was PCR-amplified with primer pair Prickle_kozak_Nhe1 and Prickle1_-T2A_Hd3Pml1. mCherry-NLS (nuclear localization signal) was amplified with the primer pair T2A_Hd3Pml1_Fw and Rb_bGlob_pA_Sac1 using CAGGS-EGFP-T2A-Cherry-NLS (C61) as template. Prickle1-T2A-mCherry-NLS was amplified with Prickle_kozak_Nhe1 and Rb_bGlob_pA_Sac1 and cloned into NheI/ SacI site of CAGGS-EGFP-loxP-Cherry (C36). Full-length axolotl *Vangl2* was cloned by PCR from day 6 tail blastema cDNA with primer pairs Vangl2_fullFw1 and Vangl2_fullRv1 and cloned into pCR2.1 (Invitrogen). For CAGGS-Vangl2 construct, *Vangl2* ORF was PCR-amplified with primer pairs Vangl2_kozak_Xba1 and Vangl2_stopER1 and cloned into NheI/EcoRI site of CAGGS-mCherry. Primer sequences can be found in *Supplementary file 2*.

## Electroporation

Electroporation of the spinal cord was carried out as previously described (*Albors and Tanaka, 2015*). Briefly, 1 µl of DNA solution (1 µg/µL) combined with Fast Green was microinjected into the spinal cord central canal using borosilicate glass capillary needles. Plasmids used were Prickle1-T2A-Cherry-NLS and CAGGs-Vangl2 co-electroporated with CAGGs-mCherry-NLS to assess electroporation efficiency. In control axolotls, CAGGs-mCherry-NLS was used alone. Electroporation conditions were 70V 5 ms length with 50 ms interval (total two pulses) followed by four bipolar pulses of 40 V 50 ms with 1 s interval and 10% voltage decay after each pulse (total of eight pulses), using a NEPA21 electroporator. Electroporated axolotls were placed in water cups and allowed to recover. 48 hours later, mCherry fluorescence was used to assess electroporation efficiency. Axolotls with less than ~60% of spinal cord tissue electroporated or damaged tails were discarded. Axolotls with successfully electroporated spinal cords were amputated, placed back in their water cups, and allowed to regenerate.

## Spinal cord outgrowth measurements

Images of regenerating tails were acquired on an Olympus SZX16 stereomicroscope using cellF software. The spinal cord outgrowth was measured from bright field images by drawing a line from the amputation plane to the tip of the regenerating spinal cord using the segmented line tool in Fiji.

## Cell division orientation analysis

To analyse the orientation of cell divisions, consecutive 50-µm cross-sections of the posterior-most portion of tails 3-, 6-, and 14-days after tail amputation were collected. Next, slides were washed in PBS, permeabilized with 0.3% PBS/Triton X-100 and nuclei stained with Hoechst overnight at 4°C. Series of optical sections of late anaphase and telophase cells, in which the position of the mitotic spindle is set (*Adams, 1996*), were collected throughout the length of the spinal cord with a Zeiss C-Apochromat 63x 1.2W objective on a Zeiss LSM 510 confocal microscope. Z-stacks of 1-µm z-steps were capture for each cell.

The orientation of the mitotic spindle was used as a proxy for the orientation of a cell division. To test whether cell divisions were oriented along the AP axis the projection of the mitotic spindle on the AP axis ($Proj^S_{AP}$) was determined as the scalar product between a vector indicating the orientation of the mitotic spindle ($S$) and a vector indicating the AP axis ($AP$):

$$Proj^S_{AP} = S \cdot AP.$$

The mitotic spindle poles were deduced from the lack of DNA staining in the pair of sister chromatids (*Figure 4A*). The mitotic spindle vector ($S$) was calculated from the coordinates of both spindle poles ($A = (x_A, y_A, z_A)$ and $B = (x_B, y_B, z_B)$):

$$S = \frac{B - A}{||B - A||},$$

and similarly, the vector indicating the AP axis (*AP*) was determined by the two points $C = (x_C, y_C, z_C)$ and $D = (x_D, y_D, z_D)$ located in the first and last optical section of each z-stack, in the middle of the central canal of the spinal cord:

$$AP = \frac{D - C}{||D - C||}$$

The value of the projection $Proj^S_{AP}$ ranges from 0, for divisions orthogonal to the AP axis, to 1, for divisions parallel to the AP axis (*Figure 4C*). To test whether cell divisions were significantly oriented along the AP axis or randomly oriented the following procedure was performed: for randomly oriented cell divisions the projections on the AP axis are uniformly distributed between 0 and 1 with a mean value of 0.5 and standard deviation $1/\sqrt{12}$. For a sample size of *N* cells, the observed mean projection is expected to be found in the interval $\left[0, 0.5 + 1.96/\sqrt{12N}\right]$ with probability 0.975. Hence, cell divisions in a specific region were considered oriented if the mean $Proj^S_{AP}$ value was higher than $0.5 + 1.96/\sqrt{12N}$ (p<0.025).

## Cleavage plane orientation analysis

Whole-mount *Vangl2*-overexpressing and control spinal cords at day 4 were immunostained for PH3, to identify mitotic cells, and GFAP, to visualize the apical surface of the spinal cord. Hoechst was used to label DNA. Images were taken with a Zeiss C-Apochromat 40x 1.2W objective on a Zeiss LSM 780. Late anaphase or telophase cells, with a set cleavage plane (*Adams, 1996*), were considered for analysis. To accurately estimate cleavage planes, only sister chromatids that appear of similar size and shape in at least 3 consecutive optical sections, and thus were perpendicular to the plane (z) of scanning, were considered further for analysis. A line through the spindle poles, inferred from the lack of DNA staining in the sister chromatids, was drawn as the mitotic spindle. The orientation of cleavage plane was determined by measuring the angle between the line orthogonal to the mitotic spindle and the apical surface of the spinal cord with the angle tool in Fiji.

## EdU labeling

For EdU pulse labeling (*Figure 5—figure supplement 2*), axolotls received a single intraperitoneal injection of EdU (0.5 mg/ml in PBS) and their tails were collected 2 hours later. For labelling proliferating cells and their progeny (*Figure 1H*, *Figure 8*), axolotls were injected EdU daily for 6 days (*Figure 1H* and *Figure 8*) and their tails were collected 1 hour after the last injection. EdU was detected on cross-sections using Click-iT EdU Alexa Fluor 488 Imaging Kit (Life Technologies) following manufacturer's instructions. Antibody immunostaining was carried out after EdU detection as described above.

## Newborn neuron counts

Every third cross-section across the entire regenerating spinal cord and at least 10 cross-sections from non-regenerating regions were imaged. Single optical sections were taken from the middle of the tissue sections with a Zeiss C-Apochromat 40x 1.2W objective on a Zeiss LSM 780. For *Figure 1H*, the number of SOX2[+]EdU[+] cells and total number of EdU[+] cells per cross-section were counted to calculate the fraction of EdU[+] cells that remained SOX2[+], and the fraction of EdU[+] cells that differentiated (SOX2[-]) during the EdU labeling time. For *Figure 8*, the fraction of newborn neurons during the EdU labeling time was calculated as Tuj1[+]EdU[+]/EdU[+].

## Apoptotic index

The fraction of apoptotic cells was determined by counting active caspase-3[+] cells within the regenerating portion of control and *Vangl2*-electroporated spinal cords. For that, non-consecutive cross-sections spanning the entire regenerate were immunostained for active caspase-3, to detect apoptotic cells, and Tuj1, to delineate the spinal cord tissue. Hoechst was used to label nuclei. Single optical sections were taken with a Zeiss C-Apochromat 40x 1.2W objective on a Zeiss LSM 780. The number

of active caspase-3$^+$ nuclei and the number of nuclei per cross-section were counted to calculate the apoptotic index as active caspase 3$^+$/Hoechst$^+$ for each replicate.

## Quantitative real-time PCR

The regenerating portion of control and *Prickle1*-overexpressing spinal cords were microdissected at day 6 and pooled in each of the three biological replicates per group. Total RNA was extracted using TRIzol (Life Technologies) and purified using RNeasy MinElute cleanup kit (QIAGEN). Power SYBR Green QPCR Master Mix (Life technologies) was used to carry out qPCRs in a LightCycler 480 thermal cycler (Roche). LightCycler 480 software (Roche) was used to analyze relative gene expression. Relative *Prickle1* expression was normalized to mRNA levels of the housekeeping gene *Rpl4*. The primers used were for *Prickle1*: forward 5'-CACTGGATTTCCGACAACCT-3' and reverse 5'-CGT-ACCATGGTGTCGACTTG-3', and for *Rpl4*: forward 5'-CACAAGATGACAAATACAGACCTTG-3' and reverse 5'-CATATGGGTTCAGTTTTACCATGAC-3'.

## Statistics

The statistical analysis carried out to test the statistical significance of each dataset is indicated in the figure legends. Mean, s.d., s.e.m., Student's *t*-test, and Mann-Whitney U test were carried out using GraphPad Prism 6 (GraphPad Software, www.graphpad.com). NanoString data analysis was carried out using the DESeq package in R as described above. *P* values < 0.05 were considered statistically significant. No statistical method was used to determine sample size.

## Acknowledgements

We are grateful to Beate Gruhl, Sabine Mögel, Anja Wagner, and Heino Andreas for outstanding axolotl care. We thank Eugen Nacu, Tatiana Sandoval-Guzmán, and Josh Currie for comments on this manuscript, and all the members of the Tanaka lab for stimulating discussions. This work was supported by grants from the Human Frontier Science Program (HFSP) RGP0016/2010, DFG-274/2-3/SFB655 'Cells into tissues', and the Center for Regenerative Therapies to E.M.T. and Agencia Nacional de Promoción Científica y Tecnológica (ANPCyT) PICT-2014-3469 to O.C. A.R.A. was funded by a DIGS-BB fellowship; F.R. and O.C. were funded by the HFSP and the German Ministry for Education and Research (BMBF, grant 0316169A). O.C. is a career researcher from Consejo Nacional de Investigaciones Científicas y Técnicas (CONICET) of Argentina.

## Additional information

### Funding

| Funder | Grant reference number | Author |
|---|---|---|
| Deutsche Forschungsgemeinschaft | DFG-274/2-3/SFB655 | Aida Rodrigo Albors<br>Akira Tazaki<br>Sergej Nowoshilow<br>Elly M Tanaka |
| Human Frontier Science Program | RGP0016/2010 | Aida Rodrigo Albors<br>Fabian Rost<br>Osvaldo Chara<br>Elly M Tanaka |
| Center for Regenerative Therapies Dresden | | Aida Rodrigo Albors<br>Akira Tazaki<br>Sergej Nowoshilow<br>Elly M Tanaka |
| DIGS-BB Program | PhD Fellowship | Aida Rodrigo Albors |
| Agencia Nacional de Promoción Científica y Tecnológica | PICT-2014-3469 | Osvaldo Chara |
| Bundesministerium für Bildung und Forschung | 0316169A | Fabian Rost<br>Osvaldo Chara |

The funders had no role in study design, data collection and interpretation, or the decision to submit the work for publication.

## Author contributions
ARA, AT, EMT, Conception and design, Acquisition of data, Analysis and interpretation of data, Drafting or revising the article; FR, OC, Analysis and interpretation of data, Drafting or revising the article; SN, Drafting or revising the article, Contributed unpublished essential data or reagents

## Author ORCIDs
Aida Rodrigo Albors, http://orcid.org/0000-0002-9573-2639
Akira Tazaki, http://orcid.org/0000-0002-7151-0089
Fabian Rost, http://orcid.org/0000-0001-6466-2589

## Ethics
Animal experimentation: The axolotl animal work was performed under permission granted in animal license number DD24-9168.11-1/012-13 conferred by the Animal Welfare Commission of the State of Saxony (Landesdirektion, Sachsen).

# Additional files

## Supplementary files
• Supplementary file 1. Cell cycle parameters of uninjured, non-regenerating, and regenerating neural stem cell populations in the axolotl spinal cord. The numbers indicate the average length (in hours) of the total cell cycle (TC), G1 phase (TG1), S phase (TS), G2 phase (TG2), and M phase (M), extracted from the cumulative BrdU data using a modeling approach (see Materials and methods). Errors, 1σ confidence intervals.

• Supplementary file 3. List of genes and target sequences used for NanoString gene expression analysis.

• Supplementary file 2. Primer sequences used for plasmid construction in this study.

## Major datasets
The following previously published dataset was used:

| Author(s) | Year | Dataset title | Dataset URL | Database, license, and accessibility information |
| --- | --- | --- | --- | --- |
| Olivera-Martinez I, Schurch N, Li RA, Song J, Halley PA, Das RM, Burt DW, Barton GJ, Storey KG | 2014 | Transcription profiling during differentiation of the neural tube in HH stage 10 chick embryos | https://www.ebi.ac.uk/arrayexpress/experiments/E-MTAB-2734/ | Publicly available at ArrayExpress (accession no. E-MTAB-2734) |

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
