## [Decision Letter]

Thank you for submitting your work entitled "Planar cell polarity-mediated induction of neural stem cell expansion during axolotl spinal cord regeneration" for peer review at *eLife*. Your submission has been favorably evaluated by Janet Rossant (Senior editor), a Reviewing editor, and two reviewers.

The reviewers have discussed the reviews with one another and the Reviewing editor has drafted this decision to help you prepare a revised submission.

Summary:

The authors uncover key molecular mechanisms underlying spinal cord regeneration in the axolotl. The work identifies non-canonical Wnt/PCP signaling as a key player in this process, including transcriptional up-regulation of mediators of this pathway and its requirement for a switch from neuron generating to progenitor generating divisions in the regenerating neural tube. The authors further show that this correlates with the ability of PCP signaling to influence mitotic spindle orientation and so identify a cell biological explanation for this change in cell behaviour. The authors take advantage of some interesting tools; the paper is well written, and the data robust. These findings are interesting and might be important for understanding the spinal cord regeneration mechanisms in general.

The authors have tackled the problem of the lack of an axolotl reference genome for RNA seq studies by using recent expression profiling data from the chick embryonic spinal cord to identify gene cohorts likely to be associated with distinct steps in the neural differentiation process. The latter data could be analysed and discussed further in this work, data on lack of cell death following manipulation of PCP signaling should also be provided.

Essential revisions:

1) In the Introduction, can the authors clarify what is meant by a neural stem cell in this context? They should mention that the axolotl spinal cord produces neurons by asymmetric division of neural progenitors throughout life.

2) The comparison with data sets from Olivera-Martinez et al. 2014 should be referred to accurately. These authors pooled in silico, data from SZ and PNT and from CNT and RNT, (rather than SZ and PNT vs NT). Is this the specific comparison used for generating the list of 100 genes?

3) It would be helpful to indicate which genes are significantly different from day 0 in Figure 1. The data presented in this figure is not dealt with in any detail in the Results or Discussion, which focus largely on PCP signaling. Can the authors provide a more detailed analysis of their data and its comparison with the chick data? For example, what differences are there between the major signaling pathways active in the PNT vs the regenerating axolotl? It appears that retinoid signaling, which is low in SZ and PNT is also declining in the regenerating spinal cord, based on readout of retinoid activity, *Rarb* and *Crabp2* levels? This would be consistent with decline in neurogenic genes known to depend on such signaling. It might also relate to the lack of change/detection of FGF signaling – an important role of which in the embryonic spinal cord is to inhibit RA signaling, if RA declines in any case this may obviate the need for increased FGF (although it is interesting that the MAPK regulator *Rasgrp3* is elevated). Can the authors comment on increased *Cdx4*, but lack of change of *Wnt8a* expression? Or the potential roles of other highly upregulated genes such as *Epha1* and *Greb1*? It would also be informative to provide a list of genes that are unchanged in comparison to those in chick PNT.

4) Is it possible to determine whether neurons re-enter the cell cycle or if it is just slow cycling neural progenitors that contribute to regeneration? For example, could neurons be labeled with fluorescent protein before amputation? This is not a necessary experiment, but it would be helpful if the authors could clarify the potential origin(s) of regenerating cells.

5) *Prickle1* expression in Figure 1 and Figure 3 is not confined to neural tissue in the caudal-most region – what is the reproducibility of this finding and does this imply a wider role for non-canonical Wnts in tail regeneration?

6) The data showing (quantifying) that lack of cellular blebbing or pyknotic nuclei needs to be provided.

7) In the Discussion can the authors comment on whether it is likely that a similar PCP role exists in zebrafish in central canal cells in response to injury and in the mouse central canal? Related to this, it would be helpful to make clear differences that might be due lesion vs amputation conditions.

8) Can they also comment on *Rarb/Crabp2* expression as this might be expected when neurogenic genes are downregulated. Does this relate to what is happening in adjacent tissues? Which might supply RA or stop doing so?

9) Figure 4 and Figure 5; It seems somewhat contradictory that upregulation of *Prickle1* in the regenerative zone leads to the AP division phenotype, yet overexpression of *Prickle1* negates the AP oriented divisions. If possible, it would be worth confirming that the expression levels are far higher in the overexpression phenotype relative to the native regeneration induced expression. More directly, shRNA/morpholino experiments would be more helpful to verify the dependence of the AP oriented divisions on *Prickle1* overexpression.

10) It is also important to verify (or cite references if previously determined) that *Prickle1* overexpression does not alter cell cycle kinetics reported for the control condition in Figure 2.

11) Similarly, for Vangl2 it would be important to examine alterations of the cell cycle.

The authors state: "Vangl2 has a non-cell autonomous effect, causing polarity defects not only in overexpressing cells but also in their neighbors" The polarity effects here need to be further explained especially in relation to the data presented how these relate to the results presented in Figure 4 and Figure 5. If such experiments haven't been previously carried out using the methodology presented here, they should be performed.

12) It seems possible that the lack of tissue regeneration shown with Vangl2 overexpression could be due to a migration deficit. Experiments should be performed to address this.

---

## [Author Response]

Essential revisions:

1) In the Introduction, can the authors clarify what is meant by a neural stem cell in this context? They should mention that the axolotl spinal cord produces neurons by asymmetric division of neural progenitors throughout life.

We thank the reviewer for the tip. We have now included the sentence:

“Cells with stem cell features persist throughout life lining the spinal cord central canal of the axolotl. These cells divide occasionally to add neurons to the intact central nervous system of the axolotl, presumably by asymmetric divisions (Holder et al., 1991)”.

Furthermore, to expand the explanation of the neural stem cell features, we have now included the sentences:

“Importantly, lineage tracing of cells lining the central canal showed that at least some cells in the source zone become highly proliferative and multipotent, populating different dorsal-ventral regions of the spinal cord (McHedlishvili et al., 2007). Furthermore clonal isolates, when transplanted into the regenerating spinal cord contributed to neurons, astrocytes, as well as oligodendrocytes in all regions of the spinal cord, indicating that a single cell can give rise to all cells of the spinal cord (McHedlishvili et al., 2012).”

2) The comparison with data sets from Olivera-Martinez et al. 2014 should be referred to accurately. These authors pooled in silico, data from SZ and PNT and from CNT and RNT, (rather than SZ and PNT vs NT). Is this the specific comparison used for generating the list of 100 genes?

Indeed, we generated the 100-gene list from the pooled SZ+PNT versus CNT+RNT comparison. We have now referred to this accurately throughout the text.

3) It would be helpful to indicate which genes are significantly different from day 0 in Figure 1 diagram.

Genes significantly different from day 0 are already highlighted in the heat map (Figure 1), as we explain in the figure legend. The first row shows the day 1 versus day 0 comparison; the second row day 6 versus day 0 comparison; and the third row shows the chick dataset, SZ+PNT versus CNT+RNT. Significantly regulated genes are colored blue, when downregulated, and red, when upregulated in day 1 or day 6 samples compared to day 0. Genes that do not change or are not expressed are colored grey. The data used to generate the heat map is also available as [Supplementary-material SD1-data]. Previously we only referred to this dataset in the figure legend this but we have now added the reference in the main text, and hope that it is easier to find.

The data presented in this figure is not dealt with in any detail in the Results or Discussion, which focus largely on PCP signaling. Can the authors provide a more detailed analysis of their data and its comparison with the chick data? For example, what differences are there between the major signaling pathways active in the PNT vs the regenerating axolotl?

We agree that the description of this very interesting dataset had been a bit sparse. We have now expanded the section describing the genes found in this dataset in the Results, subheading “Neural stem cells in the injured axolotl spinal cord reactivate molecular programs associated with embryonic neuroepithelial cells”.

*It appears that retinoid signaling, which is low in SZ and PNT is also declining in the regenerating spinal cord, based on readout of retinoid activity,* Rarb *and* Crabp2 *levels? This would be consistent with decline in neurogenic genes known to depend on such signaling. It might also relate to the lack of change/detection of FGF signaling – an important role of which in the embryonic spinal cord is to inhibit RA signaling, if RA declines in any case this may obviate the need for increased FGF (although it is interesting that the MAPK regulator* Rasgrp3 *is elevated).*

That is a very interesting observation from the reviewers. Indeed, the downregulation of RA target genes *Rarb, Crabp1* and *Meis2* suggests that retinoid signalling declines during regeneration. We have now commented on this change in the Results (from paragraph three). However, regarding FGF signaling, it is important to bear in mind that we only looked at the expression changes of a limited, 100-gene set. Thus, although our data show that Fgf8 is not expressed, we cannot rule out that other FGF ligands such as FGF2 are responsible for the decline of RA signaling or the activation of the MAPK pathway. Zhang and colleagues actually showed that FGF2 is upregulated in the regenerating spinal cord in newts (PMID:10985857). Interestingly, they also showed dynamic changes in the expression of FGF receptors in the regenerating spinal cord. Consistent with our gene expression analysis in axolotl, FGFR2 is downregulated in the early stages of newt spinal cord regeneration –while FGFR1 is upregulated (PMID: 12379240).

*Can the authors comment on increased* Cdx4*, but lack of change of* Wnt8a *expression?*

As discussed above, the upregulation of *Cdx4* could be triggered by Wnt ligands not included in our 100-gene set such as *Wnt3a* (see, for instance, PMID: 16309666). Another possibility is that FGF/MAPK signaling is regulating *Cdx4* expression (PMID: 16982047).

*Or the potential roles of other highly upregulated genes such as* Epha1 *and* Greb1*?*

*Greb1* has been associated with estrogen-induced cell proliferation and cancer progression (PMID: 15986123) and *Epha1* was first isolated from cancer cell lines (PMID: 2825356). It is thus likely that both contribute to the conversion to the more proliferative state of regenerating neural stem cells in axolotl. We have now included a few notes about these highly upregulated genes in the Results (paragraphs two and four).

It would also be informative to provide a list of genes that are unchanged in comparison to those in chick PNT.

This information can be also found in [Supplementary-material SD1-data].

4) Is it possible to determine whether neurons re-enter the cell cycle or if it is just slow cycling neural progenitors that contribute to regeneration? For example, could neurons be labeled with fluorescent protein before amputation? This is not a necessary experiment, but it would be helpful if the authors could clarify the potential origin(s) of regenerating cells.

The contribution of neurons to the regenerating spinal cord has already been addressed by Zhang and colleagues (PMID: 12557212). They retro-labeled motor neurons using fluorescent dextran before amputation and followed them during regeneration to find that some neurons move into the regenerating spinal cord, but do not dedifferentiate or re-enter the cell cycle. It is well accepted that the new spinal cord stems mainly from the SOX2^+^ neural stem cell population (PMID: 17507409 and 12557212). However, we have added a small note in the Introduction to make this point clear (paragraph two).

5) Prickle1 expression in Figure 1 and Figure 3 is not confined to neural tissue in the caudal-most region – what is the reproducibility of this finding and does this imply a wider role for non-canonical Wnts in tail regeneration?

Indeed, *Prickle1* and *Wnt5a* are consistently upregulated in the blastema/mesenchymal cells surrounding the spinal cord. Based on previously literature it is likely that PCP also directs polarized growth/morphogenesis in the mesenchymal component of developing and regenerating tissues. Sugiura and colleagues showed in *Xenopus laevis* that provision of *Xwnt-5a* in lateral sites of injured tadpole tails can induce regeneration of a complete ectopic tails containing both spinal cord and mesodermal tissues (PMID 18977433). In addition, Gros and colleagues (PMID 21055947) showed a role for *wnt5* signalling in controlling cell-oriented behaviors of limb bud mesenchyme leading to directional during limb bud outgrowth. These observations indeed indicate that non-canonical Wnt signalling acts also in mesenchymal tissues to direct oriented outgrowth of vertebrate appendages.

6) The data showing (quantifying) that lack of cellular blebbing or pyknotic nuclei needs to be provided.

We thank the reviewers for the suggestions. We have now quantified the fraction of apoptotic cells in control and Vangl2-overexpressing spinal cords and show this in Figure 7—figure supplement 1.

7) In the Discussion can the authors comment on whether it is likely that a similar PCP role exists in zebrafish in central canal cells in response to injury and in the mouse central canal? Related to this, it would be helpful to make clear differences that might be due lesion vs amputation conditions.

We have now added an extensive section at the end of the Discussion to reflect on the function of Wnt5 in developmental and regenerative contexts. The work in development clearly shows that Wnt5 expression is associated with outgrowths of both epithelium and mesenchyme-containing structures. The functional analysis of Wnt5 in different regeneration systems has not yielded as uniform of a conclusion. While in amphibians Wnt5 is clearly associated with tail outgrowth (for example this study and PMID: 18977433), in the zebrafish, fin activation of Wnt5 supressed regeneration (PMID: 17185322). It is unclear if this latter observation is due to the global activation of Wnt5 signalling, which would derange PCP making regeneration less directed, or whether, as hypothesize as the authors, this reflects an antagonism between canonical and non-canonical Wnt pathways in fin cells. To our knowledge, the role of Wnt/PCP signalling after spinal cord injury has not been studied in zebrafish or mouse, but would clearly be important for comparative purposes.

*8) Can they also comment on* Rarb/Crabp2 *expression as this might be expected when neurogenic genes are downregulated. Does this relate to what is happening in adjacent tissues? Which might supply RA or stop doing so?*

We now provide a discussion of this expression in the Results as mentioned above, and discuss this issue in the Discussion, starting in paragraph five. Indeed, it is likely that non-spinal cord cell sources provide retinoic acid signalling dynamically during the course of regeneration and, given the similarities between the developmental and regeneration situations, that at one point, RA is a pro-neurogenic factor during regeneration. In another project in the laboratory we are starting to study these issues with respect to myotome regeneration, and therefore eventually we will link the differentiation of myotomal tissues with the onset of neurogenesis in the spinal cord. The focus of the current manuscript is primarily the early switch from the uninjured state to the PNT-like state. Taking all into consideration, an experimental link between RA from non-spinal cord sources and neurogenesis will be taken up in a future manuscript.

*9) Figure 4 and Figure 5; It seems somewhat contradictory that upregulation of* Prickle1 *in the regenerative zone leads to the AP division phenotype, yet overexpression of* Prickle1 *negates the AP oriented divisions. If possible, it would be worth confirming that the expression levels are far higher in the overexpression phenotype relative to the native regeneration induced expression.*

We thank the reviewers for pointing out this difficulty of understanding the rational to this experiment. To address this issue, we have now inserted background information with references that validates the use of overexpression to disrupt PCP (please refer to paragraph two, subheading “PCP is required for oriented cell divisions in the regenerating spinal cord”). We have also quantified the expression levels of *Prickle1* in control and *Prickle1*-overexpressing spinal cord and added this in Figure 5—figure supplement 1.

More directly, shRNA/morpholino experiments would be more helpful to verify the dependence of the AP oriented divisions on Prickle1 overexpression.

Unfortunately GeneTools were unable to design an acceptable morpholino against the axolotl *Prickle1* 5’ sequence and splicing-disrupting morpholinos are not realistic due to lack of genome sequence. Our laboratory is currently working on other (CRISPR/Cas9) knockdown strategies for the spinal cord, but validation of these technical developments will take time beyond the revision scope of this manuscript.

10) It is also important to verify (or cite references if previously determined) that Prickle1 overexpression does not alter cell cycle kinetics reported for the control condition in Figure 2.

We have now addressed this issue by assessing the rate of EdU incorporation in control and *Prickle1*-overexpressing cells, and found that it is indistinguishable from mCherry-only-transfected cells. The data is in Figure 5—figure supplement 2. We have chosen EdU incorporation as a measure for cell cycle kinetics due to the relatively few Prickle-overexpressing cells we can obtain with the very large *Prickle1* construct.

11) Similarly, for Vangl2 it would be important to examine alterations of the cell cycle.

In Figure 7 we had already quantitated the frequency of mitotic figures in Vangl2-expressing versus control expressing spinal cords, which indicates that the cycling characteristics of the Vangl2 expressing samples. It should be noted that we have analyzed the Vangl2 overexpression phenotype at the tissue level because the overexpression of Vangl2 has a non-cell autonomous effect on PCP in other vertebrate contexts, as we describe in paragraph two, subsection “PCP is required for oriented cell divisions in the regenerating spinal cord”.

The authors state: "Vangl2 has a non-cell autonomous effect, causing polarity defects not only in overexpressing cells but also in their neighbors" The polarity effects here need to be further explained especially in relation to the data presented how these relate to the results presented in Figure 4 and Figure 5. If such experiments haven't been previously carried out using the methodology presented here, they should be performed.

We have now included more extensive background information on the PCP system and the different classes of phenotypes (autonomous versus non-autonomous) that are expected based on which core component is overexpressed (please see paragraph two of the subsection entitled “PCP is required for oriented cell divisions in the regenerating spinal cord”).

12) It seems possible that the lack of tissue regeneration shown with Vangl2 overexpression could be due to a migration deficit. Experiments should be performed to address this.

In the lab, we are currently performing two other projects that impinge on this question. First we are performing live time-lapse imaging of regeneration. We observe that the major time of what could be termed collective cell migration occurs at 48 hours, which is the phase before the onset of the Vangl2 phenotypes. At this stage, Vangl2 overexpressing spinal cords extend rather normally. In a second project in the laboratory, we have followed labeled cells or small clusters during spinal cord regeneration quantitatively based on their initial position behind the amputation plane. This data, along with some of the data presented in the current manuscript, are being used to construct a quantitative model of regeneration. The results so far indicate that migration plays a minor role in spinal cord regeneration, and that growth can be explained largely by proliferation and some slight “pushing” from more anterior cells, due to their slow expansion. This is an extensive project and will not be submitted prior to the current manuscript. We would prefer to keep that work separate, but are happy to provide the data for perusal when requested.